# Neighbour-Driven Gaussian Process Variational Autoencoders for Scalable Structured Latent Modelling

**Xinxing Shi** [1]   **Xiaoyu Jiang** [1]   **Mauricio A. Álvarez** [1]

## Abstract

Gaussian Process (GP) Variational Autoencoders (VAEs) extend standard VAEs by replacing the fully factorised Gaussian prior with a GP prior, thereby capturing richer correlations among latent variables. However, performing exact GP inference in large-scale GPVAEs is computationally prohibitive, often forcing existing approaches to rely on restrictive kernel assumptions or large sets of inducing points. In this work, we propose a neighbour-driven approximation strategy that exploits local adjacencies in the latent space to achieve scalable GPVAE inference. By confining computations to the nearest neighbours of each data point, our method preserves essential latent dependencies, allowing more flexible kernel choices and mitigating the need for numerous inducing points. Through extensive experiments on tasks including representation learning, data imputation, and conditional generation, we demonstrate that our approach outperforms other GPVAE variants in both predictive performance and computational efficiency.

## 1. Introduction

Variational Autoencoders (VAEs) (Kingma & Welling, 2014) have achieved remarkable success in a variety of tasks ranging from representation learning (Higgins et al., 2017; Kim & Mnih, 2018) to generative modelling (Sønderby et al., 2016; Razavi et al., 2019). While a conventional VAE assumes a fully factorised Gaussian prior over the latent variables to enable amortised inference, this assumption can be overly restrictive in many real-world scenarios involving

[1]Department of Computer Science, University of Manchester, Manchester, UK. Correspondence to: Xinxing Shi <xinxing.shi@postgrad.manchester.ac.uk>, Xiaoyu Jiang <xiaoyu.jiang@postgrad.manchester.ac.uk>, Mauricio A. Álvarez <mauricio.alvarezlopez@manchester.ac.uk>.

*Proceedings of the $42^{nd}$ International Conference on Machine Learning*, Vancouver, Canada. PMLR 267, 2025. Copyright 2025 by the author(s).

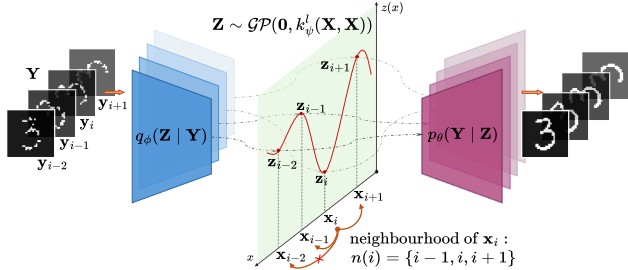

**Figure 1:** GPVAE places GP prior over the latent variables $\mathbf{Z}$ to model correlations in the structured data $(\mathbf{x}_i, \mathbf{y}_i)$. Our approach is to approximate the dense GP prior by leveraging the associated neighbourhood $(\mathbf{x}_{n(i)}, \mathbf{y}_{n(i)})$.

sequential, spatial, or other structured data. In these cases, it is crucial to model correlations among latent variables.

A natural way to introduce such correlations is to adopt a Gaussian Process (GP) prior over the latent representations, resulting in Gaussian Process Variational Autoencoders (GP-VAEs) (Casale et al., 2018). By encoding the latent variables as realisations from a GP, GPVAEs enforce structured dependencies through kernels (as shown in Fig. 1). Unfortunately, the direct application of GPs leads to an $\mathcal{O}(N^3)$ computational overhead ($N$ is the number of training samples), making naive approaches prohibitively expensive in large-scale settings.

To address this limitation, one line of research explores specific kernel structures, such as certain Matérn families (Zhu et al., 2023) and low-rank kernels (Casale et al., 2018). While effective in addressing scalability challenges, these approaches are constrained by their narrow kernel choices. Other scalable GPVAE variants have been proposed, notably using *inducing points* (Ashman et al., 2020; Jazbec et al., 2021), a small set of pseudo-points that approximates the GP posterior. These methods, however, may require numerous inducing points in rapidly varying data or encounter difficulties in optimising the inducing points (Wu et al., 2022). Moreover, fully Bayesian extensions using sampling-based methods offer good uncertainty calibration but often require longer runtime for sufficient samples (Tran et al., 2023).

In this work, we propose an efficient neighbour-driven approach to GPVAE training and prediction (as illustrated in Fig. 1). Our key insight is that in many structured datasets—such as video frames with temporal adjacency or spatial regions with local patterns—focusing on a small set of nearest neighbours captures most of the essential correlation structures. This intuition aligns with the "first law of geography," which underlies various Nearest Neighbour GP (NNGP) methods (Vecchia, 1988; Stein et al., 2004). Inspired by recent evidence that NNGPs can outperform sparse GPs with inducing points (Hensman et al., 2013) in large-scale geostatistics and machine learning tasks (Datta et al., 2016b; Tran et al., 2021b; Wu et al., 2022; Cao et al., 2023; Allison et al., 2024), we introduce two latent GP prior approximations to neighbour-driven latent modelling in GPVAEs: **(1)** *Hierarchical Prior Approximation (HPA)*: enforces sparsity in dense GP covariance matrices via a hierarchical mechanism that selects local neighbourhoods in each mini-batch; **(2)** *Sparse Precision Approximation (SPA)*: chain-factorises the GP prior into conditional distributions involving *only* the nearest neighbours. Both methods support structured latent modelling with flexible kernels while avoiding large sets of inducing points or rigid kernel assumptions. By integrating these approximations into mini-batch amortised inference, our approach leverages local correlation principles for scalable GPVAE inference, preserving essential latent structures with minimal overhead.

**Contributions** This paper makes the following contributions: **(1)** We introduce two neighbour-driven GP prior approximations into GPVAE inference that capture essential latent structure while remaining scalable. **(2)** We present an amortised training procedure that jointly learns the encoder, decoder, and kernel parameters in a mini-batch fashion, accommodating a wide range of kernels without restrictive assumptions or a large number of inducing points. **(3)** Our framework delivers competitive performance on multiple tasks for latent representation learning, data imputation, and conditional generation. Empirical experiments demonstrate that our approach improves both predictive accuracy and training speed compared to existing GPVAE baselines.

## 2. Background

Section 2.1 revisits GPVAEs and the scalability bottleneck from the full GP prior. Section 2.2 summarises the widely used inducing-point formulation for scalable GPVAE inference, while Section 2.3 reviews NNGP approximations that motivate our approach.

### 2.1. Gaussian Process Variational Autoencoder

A GP $f(\mathbf{x}) \sim \mathcal{GP}(m(\mathbf{x}), k(\mathbf{x}, \mathbf{x}'))$ is a stochastic process for which any finite subset of random variables follows a joint Gaussian distribution (Rasmussen & Williams, 2006). The covariance matrix of that joint distribution is determined by the kernel function $k : \mathbb{R}^D \times \mathbb{R}^D \to \mathbb{R}$, which encodes how sample points of the input space $\mathbf{x} \in \mathbb{R}^D$ correlate with each other.

Consider $N$ pairs of data points

$$\mathbf{Y} = [\mathbf{y}_n]_{n=1}^N \in \mathbb{R}^{N \times K}, \ \mathbf{X} = [\mathbf{x}_n]_{n=1}^N \in \mathbb{R}^{N \times D},$$

where $\mathbf{y}_n \in \mathbb{R}^K$ is the $n$-th observation and $\mathbf{x}_n \in \mathbb{R}^D$ corresponds to the auxiliary information (e.g., video timestamps or spatial coordinates). We define associated latent variables $\mathbf{Z} = [\mathbf{z}_n]_{n=1}^N \in \mathbb{R}^{N \times L}$, with $L$ latent channels. By the construction of GPVAE, each latent channel $\mathbf{z}^l \in \mathbb{R}^N$ follows $\mathbf{z}^l \sim \mathcal{N}(\mathbf{0}, k_\psi^l(\mathbf{X}, \mathbf{X}))$, where $k_\psi^l$ is the kernel with parameters $\psi$. The generative process is given by

$$p_\psi(\mathbf{Z} \mid \mathbf{X}) = \prod_{l=1}^L \mathcal{N}(\mathbf{z}^l \mid \mathbf{0}, k_\psi^l(\mathbf{X}, \mathbf{X})), \qquad (1)$$

$$p_\theta(\mathbf{Y} \mid \mathbf{Z}) = \prod_{n=1}^N p_\theta(\mathbf{y}_n \mid \mathbf{z}_n), \qquad (2)$$

where the latent prior factorises over channels and $p_\theta(\mathbf{y}_n \mid \mathbf{z}_n)$ is modelled by a decoder network with parameters $\theta$. In the subsequent sections, we will use the notation $\mathbf{K}_{\mathbf{XX}} := k(\mathbf{X}, \mathbf{X})$ to represent the covariance matrix evaluated at all pairs of inputs $\mathbf{X}$.

**Variational inference for GPVAEs** To approximate the intractable posterior $p(\mathbf{Z} \mid \mathbf{Y}, \mathbf{X})$, consider the following variational distribution as in a standard VAE (Kingma & Welling, 2014; Casale et al., 2018):

$$q_\phi(\mathbf{Z} \mid \mathbf{Y}) = \prod_{n=1}^N \mathcal{N}(\mathbf{z}_n \mid \mu_\phi(\mathbf{y}_n), \sigma_\phi^2(\mathbf{y}_n)\mathbf{I}_L), \quad (3)$$

where the mean $\mu_\phi$ and variance $\sigma_\phi^2$ are given by an encoder network with parameters $\phi$. The GPVAE parameters $\{\psi, \theta, \phi\}$ can be learned by maximising the following Evidence Lower BOund (ELBO):

$$\begin{aligned} \mathcal{L} = \ & \mathbb{E}_{q_\phi(\mathbf{Z} \mid \mathbf{Y})}\left[\log p_\theta(\mathbf{Y} \mid \mathbf{Z})\right] \\ & - \mathrm{KL}\left[q_\phi(\mathbf{Z} \mid \mathbf{Y}) \parallel p_\psi(\mathbf{Z} \mid \mathbf{X})\right], \end{aligned} \qquad (4)$$

where the KL stands for Kullback–Leibler divergence. If $\mathbf{K}_{\mathbf{XX}} = \mathbf{I}_N$, the GP prior collapses to a fully factorised VAE prior and (4) recovers the standard VAE objective. In general, however, the KL term does not factorise across samples due to dense $\mathbf{K}_{\mathbf{XX}}$. As a result, (4) cannot be computed in mini-batches and incurs $\mathcal{O}(N^3)$ time complexity, making large-scale inference computationally infeasible.

## 2.2. GPVAEs Based on Inducing Points

A line of research addresses the $\mathcal{O}(N^3)$ bottleneck in GP-VAEs by adopting approximations based on inducing points (Titsias, 2009; Hensman et al., 2013; Jazbec et al., 2021). It uses an alternative variational distribution to eliminate the troublesome $p_\psi(\mathbf{Z} \mid \mathbf{X})$ in the KL term:

$$q(\mathbf{Z} \mid \mathbf{Y}, \mathbf{X}) = \prod_{l=1}^{L} \frac{p_\psi(\mathbf{z}^l \mid \mathbf{X}) \prod_{n=1}^{N} \tilde{q}_\phi(\mathbf{z}_n^l \mid \mathbf{y}_n)}{Z_{\psi,\phi}^l(\mathbf{Y}, \mathbf{X})}, \quad (5)$$

where $\tilde{q}_\phi$ is modelled by an encoder, and $Z_{\psi,\phi}^l$ is the normalising constant. Equation (5) is further approximated by following the inducing-point framework through

$$q(\mathbf{Z} \mid \mathbf{Y}, \mathbf{X}) \approx \int p(\mathbf{Z} \mid \mathbf{Z_U}) q(\mathbf{Z_U} \mid \mathbf{Y}, \mathbf{X}) \, \mathrm{d}\mathbf{Z_U},$$

where $\mathbf{U} \in \mathbb{R}^{M \times D}$ represents $M(\ll N)$ inducing locations, $\mathbf{Z_U}$ is the corresponding inducing variables, and $p(\mathbf{Z} \mid \mathbf{Z_U})$ is a conditional Gaussian. In practice, the aforementioned variational distribution $q(\mathbf{Z_U} \mid \mathbf{Y}, \mathbf{X})$ over inducing points takes a stochastic heuristic form determined by mini-batches. Additionally, the normalising constant $Z_{\psi,\phi}^l$ is subject to a lower bound associated with $\mathbf{U}$. The resulting training objective allows learning the locations of inducing points. We abbreviate the model described above to SVGPVAE, and more details are provided in Appendix D.1.

## 2.3. Nearest Neighbour Gaussian Process

NNGPs scale classical GPs by assuming that each data point depends only on a small set of nearby neighbours, typically determined by spatial or temporal distance. This locality naturally reduces computational costs: rather than dealing with a dense $N \times N$ covariance or precision matrix, NNGPs construct block-sparse or banded structures where each row or column involves only the nearest neighbours.

NNGPs have been widely used in geostatistics (Datta et al., 2016a;b; Katzfuss & Guinness, 2021; Datta, 2022). Recent works extend NNGPs to general regression tasks, employing variational approaches from inducing points (Tran et al., 2021b; Wu et al., 2022). For example, Tran et al. (2021b) present an NNGP that implements a hierarchical prior with an auxiliary random indicator to determine the selection of inducing variables. In addition, Wu et al. (2022) propose a variational GP that applies a sparse precision approximation for the inducing variable distribution. These approaches have shown advantages over the standard inducing point method, particularly on large datasets characterised by intrinsically low lengthscales, where many inducing points are required to capture local correlations effectively.

## 3. Neighbour-Driven GPVAE

Our approach adopts a similar local-neighbour perspective to NNGPs but applies it inside the latent space of GPVAEs. Rather than sparsifying GPs in the observation domain, we build scalable, neighbour-based GP priors within the latent dimensions. This adaptation brings the computational benefits of NNGPs to structured latent modelling for high-dimensional and correlated data.

In this section, we introduce two neighbour-driven GP-VAE variants that can exhibit high scalability for large-scale datasets.

### 3.1. Model Setup

We adopt the standard construction of GPVAE mentioned earlier: a multi-output GP prior on the latent variables $\mathbf{Z} = \left[\mathbf{z}^l\right]_{l=1}^{L} \in \mathbb{R}^{N \times L}$ as in (1) and a decoder network mapping each latent variable $\mathbf{z}_n \in \mathbb{R}^L$ to its observation $\mathbf{y}_n$ as in (2). Separate kernels $\{k_\psi^l\}_{l=1}^{L}$ are used across $L$ latent dimensions to fully exploit GP's expressivity. Since the latent GPs are independent, we omit the superscript for latent dimensions $l$. We also suppress the subscript for model parameters to make the notations uncluttered.

We choose to employ the standard encoder from (3) to keep the model design straightforward. Employing this conventional architecture facilitates direct comparisons with standard VAEs in our paper. As we will show in subsequent experiments, this encoder delivers competitive performance. If needed, practitioners can incorporate $\mathbf{X}$ by designing an encoder that accepts both $\mathbf{X}$ and $\mathbf{Y}$ as inputs.

### 3.2. Neighbour-Driven Inference

Our idea is to impose sparsity on the full GP prior, allowing the KL term of (4) to decompose into manageable terms while preserving essential latent correlations. To achieve this, we develop inference methods using two distinct GP prior approximations inspired by recent NNGPs.

**Hierarchical Prior Approximation (HPA)**    To formulate the inference, we introduce an additional binary random vector $\mathbf{w} \in \{0, 1\}^N \sim p(\mathbf{w})$ to indicate the inclusion of $N$ latent variables $\mathbf{Z}$. We then establish a hierarchical structure for the prior based on the selection process

$$p(\mathbf{Z} \mid \mathbf{w}) = \mathcal{N}\left(\mathbf{Z} \mid \mathbf{0}, \mathbf{D_w} \mathbf{K_{XX}} \mathbf{D_w}\right), \quad (6)$$

$$p(\mathbf{Z}, \mathbf{w}) = p(\mathbf{Z} \mid \mathbf{w}) p(\mathbf{w}), \quad (7)$$

where $\mathbf{D_w} = \mathrm{diag}(\mathbf{w})$. From (6), the correlations of the unselected variables $\mathbf{Z}_{\backslash \mathcal{I}}$ among $\mathbf{Z}$ are eliminated, where $\backslash \mathcal{I} = \{i : \mathbf{w}_i = 0\}$. Then, we set up an amortised varia-

tional distribution that uses a similar form to (6) as follows[1]

$$q(\mathbf{Z} \mid \mathbf{w}) = \mathcal{N}(\mathbf{Z} \mid \mathbf{D_w}\mu(\mathbf{Y}), \mathbf{D_w}\sigma^2(\mathbf{Y})\mathbf{D_w}), \quad (8)$$

$$q(\mathbf{Z}, \mathbf{w}) = q(\mathbf{Z} \mid \mathbf{w})p(\mathbf{w}), \quad (9)$$

where $\mu(\cdot)$ and $\sigma^2(\cdot)$ are the outputs of the encoder. By the construction of (6)-(9), the ELBO for our GPVAE is updated as

$$\mathcal{L}_{\text{HPA}} = \mathbb{E}_{p(\mathbf{w})} \big\{ \mathbb{E}_{q(\mathbf{Z}|\mathbf{w})} \log p(\mathbf{Y} \mid \mathbf{Z}) \\ -\text{KL}\left[q(\mathbf{Z} \mid \mathbf{w}) \mid\mid p(\mathbf{Z} \mid \mathbf{w})\right] \big\},$$

where the first term is the expected log-likelihood. $p(\mathbf{w})$ specifies how the variables are selected and can be any implicit distribution that we can draw samples from. In this work, sampling from $p(\mathbf{w})$ is materialised by the following neighbour-driven strategy: for each training point $(\mathbf{x}_i, \mathbf{y}_i)$ in a random mini-batch $\{(\mathbf{x}_i, \mathbf{y}_i)\}_{i \in \mathcal{I}}$, we pick up the top-$H$ nearest locations to the auxiliary point $\mathbf{x}_i$ in $\mathbf{X}$, whose indices are denoted as $n(i)$. For stochastic mini-batch training, we estimate the ELBO under the hierarchical prior by

$$\mathcal{L}_{\text{HPA}} \approx \frac{N}{|\mathcal{I}|} \sum_{i \in \mathcal{I}} \bigg\{ \mathbb{E}_{q(\mathbf{z}_i|\mathbf{y}_i)} \left[\log p(\mathbf{y}_i \mid \mathbf{z}_i)\right] \\ -\frac{1}{N}\text{KL}\left[q(\mathbf{Z}_{n(i)}) \mid\mid p(\mathbf{Z}_{n(i)})\right] \bigg\}, \quad (10)$$

where the KL divergence breaks down into several terms that involve lower-dimensional covariance matrices (i.e., $H \times H$). The original full-batch EBLO (4) can be recovered when all elements of $\mathbf{w}$ are set to one or equivalently, $H$ are set to include all $N$ training points. We refer to our model trained through (10) as GPVAE-HPA. For a detailed derivation, please see Appendix B.1.

**Sparse Precision Approximation (SPA)**  The inference in this section employs a sparse precision matrix approximation (Vecchia, 1988; Datta, 2022) to latent GPs. Consider factorising the joint distribution $p(\mathbf{Z})$ of a GP by the probability chain rule subject to some ordering

$$p(\mathbf{Z}) = p(\mathbf{z}_1) \prod_{j=2}^{N} p(\mathbf{z}_j \mid \mathbf{z}_{1:j-1}), \quad (11)$$

where $\mathbf{z}_{1:j-1}$ stands for the collection $\{\mathbf{z}_h\}_{h=1}^{j-1}$. The SPA is built by imposing conditional independence on (11) by only considering nearest neighbours, leading to

$$p(\mathbf{Z}) \approx p(\mathbf{z}_1) \prod_{j=2}^{N} p(\mathbf{z}_j \mid \mathbf{z}_{n(j)}), \quad (12)$$

where we abuse the notation $n(j)$ to represent the indices of $H$ nearest neighbours of $\mathbf{x}_j$ in $\{\mathbf{x}_h\}_{h=1}^{j-1}$ (rather than among

---

[1]We have omitted the conditioning variable $\mathbf{Y}$.

all points of $\mathbf{X}$). Proceeding with the variational distribution $q(\mathbf{Z} \mid \mathbf{Y})$ defined by (3), we obtain the ELBO based on SPA as follows

$$\mathcal{L}_{\text{SPA}} = \sum_{i=1}^{N} \mathbb{E}_{q(\mathbf{z}_i|\mathbf{y}_i)} \left[\log p(\mathbf{y}_i \mid \mathbf{z}_i)\right] \\ - \sum_{j=1}^{N} \mathbb{E}_{q(\mathbf{Z}_{n(j)})} \text{KL}\left[q(\mathbf{z}_j) \mid\mid p(\mathbf{z}_j \mid \mathbf{Z}_{n(j)})\right].$$

Therefore, the two terms of the ELBO above can be estimated by mini-batches from the data

$$\mathcal{L}_{\text{SPA}} \approx \frac{N}{|\mathcal{I}|} \sum_{i \in \mathcal{I}} \mathbb{E}_{q(\mathbf{z}_i|\mathbf{y}_i)} \left[\log p(\mathbf{y}_i \mid \mathbf{z}_i)\right] \\ - \frac{N}{|\mathcal{J}|} \sum_{j \in \mathcal{J}} \mathbb{E}_{q(\mathbf{Z}_{n(j)})} \text{KL}\left[q(\mathbf{z}_j) \mid\mid p(\mathbf{z}_j \mid \mathbf{Z}_{n(j)})\right], \quad (13)$$

where $\mathcal{I}$ and $\mathcal{J}$ are sets of mini-batch indices. We call the proposed model GPVAE-SPA since (12) essentially offers an approximation with sparse Cholesky factor for the prior precision matrix $\mathbf{K}_{\mathbf{XX}}^{-1}$ (Datta, 2022). A detailed derivation of (13) is provided in Appendix B.2. By setting $H = N$, we retain all dependencies in the probability chain in (12), thus recovering the full-batch ELBO (4) again. In contrast, setting $H = 0$ will cause the model to degenerate into conventional VAEs.

**Sparsity mechanisms and NNGP links**  HPA and SPA impose neighbour-driven sparsity in complementary ways. HPA produces a sparse *covariance* structure by "switching off" interactions between non-neighbouring latent variables through a hierarchical selection variable. In contrast, SPA factorises the GP distribution into chained conditional terms, effectively resulting in a sparse *precision* matrix. Both schemes are valid neighbour-driven approaches suitable for scalable GPVAE inference.

Conceptually, our approach closely aligns with the NNGPs by Tran et al. (2021b) and Wu et al. (2022), which enforce local sparsity on inducing variables. To draw a parallel, the proposed GPVAEs can be seen as applying such a principle to latent NNGPs with each "inducing variable" placed at each data point. We then incorporate a decoder network to model the "likelihood" $p(\mathbf{Z} \mid \mathbf{Y})$ and an encoder network for amortised inference, thereby forming our GPVAE architecture.

### 3.3. Predictive Posterior

Predictive posterior distributions $p(\mathbf{y}_* \mid \mathbf{x}_*, \mathbf{Y})$ are often used in tasks of conditional generation, where an unseen auxiliary $\mathbf{x}_* \in \mathbb{R}^D$ is given. Here, we derive an approximate predictive posterior for the proposed GPVAEs based on neighbouring latent representations. Specifically, computing

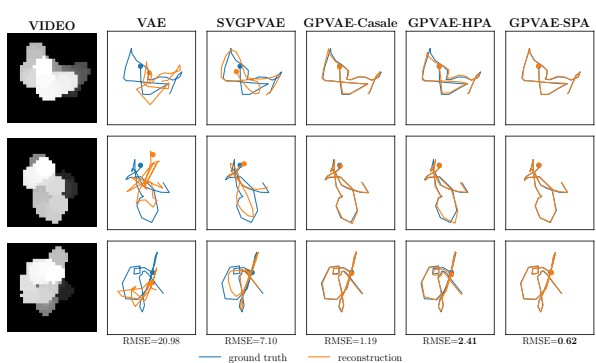

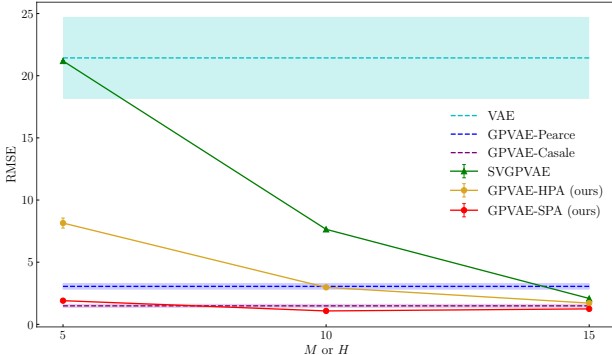

**(a)** Latent trajectory reconstruction of moving ball videos.

**(b)** Reconstruction RMSEs using different numbers of nearest neighbours/inducing points.

**Figure 2:** Latent representation learning for the moving ball dataset. **(a)** The leftmost column shows frames overlaid and shaded by time. The orange reconstructed paths are obtained using $M/H = 10$. **(b)** The results of the standard VAE and two full-batch baselines are shown with the shaded bands. The error bars and the shades indicate $\pm 1$ standard deviation.

$p(\mathbf{y}_* \mid \mathbf{x}_*, \mathbf{Y})$ at a new location $\mathbf{x}_*$ only needs to consider its $H$ nearest neighbours in $\mathbf{X}$ (denoted by the index set $n(*)$):

$$q(\mathbf{z}_* \mid \mathbf{Y}) = \int p(\mathbf{z}_* \mid \mathbf{Z}_{n(*)}) q(\mathbf{Z}_{n(*)} \mid \mathbf{Y}_{n(*)}) \, d\mathbf{Z}_{n(*)},$$

$$p(\mathbf{y}_* \mid \mathbf{x}_*, \mathbf{Y}) \approx \int p(\mathbf{y}_* \mid \mathbf{z}_*) q(\mathbf{z}_* \mid \mathbf{Y}) \, d\mathbf{z}_*,$$

where $q(\mathbf{z}_* \mid \mathbf{Y})$ is Gaussian and can be sampled efficiently in practice to give Monte Carlo estimation of $p(\mathbf{y}_* \mid \mathbf{x}_*, \mathbf{Y})$.

### 3.4. Computational Complexity

The computational complexity of our models is twofold: (1) determining the nearest neighbour structure for auxiliary data; (2) Cholesky decomposition of $H \times H$ covariance matrices in the KL term. Locating the $H$ nearest neighbours of each point takes $\mathcal{O}(HN)$ in the worst case. Our implementation leverages the Faiss package (Johnson et al., 2019) to efficiently accelerate the nearest neighbour search on GPUs. After pre-computing the nearest neighbour structure, Cholesky decomposition in (10) and (13) has complexity $\mathcal{O}(LN_bH^3)$, where $N_b$ is the mini-batch size and $L$ is the latent dimensions. Table 15 in Appendix D provides an overview of the GP complexity associated with other relevant models.

## 4. Related Work

The GPVAE was first proposed by Casale et al. (2018) to remove the i.i.d. assumption of the Gaussian prior in VAEs. The proposed GPVAE uses an encoder and a decoder as in a standard VAE, which we also adopt in our work. The work has to use a low-rank GP kernel and resort to first-order Taylor series expansion to achieve mini-batch inference, which is overly complicated and computationally inefficient.

Unlike Casale et al. (2018), Pearce (2020) and Fortuin et al. (2020) adopted alternative formulations of the variational distribution and applied GPVAE to applications of interpretable latent dynamics and time-series imputation, respectively, but their models are limited to short sequences.

As aforementioned, SVGPVAE (Jazbec et al., 2021) uses inducing points but may suffer from optimisation issues as the number of inducing points increases. LVAE (Ramchandran et al., 2021) is another inducing-point-based model restricted to additive kernels for longitudinal data with discrete instance covariates. MGPVAE (Zhu et al., 2023) exploits state-space representations of one-dimensional Matérn kernels to enable Kalman-filter-like inference. Since the inference involves both forward and backward processes, utilising parallel computation to speed up can be challenging. SGPBAE (Tran et al., 2023) treats latent variables, decoder parameters, and kernel parameters in a fully Bayesian fashion and leverages Stochastic Gradient Hamiltonian Monte Carlo (SGHMC) (Chen et al., 2014). Compared to our method, sampling from that model is usually time-consuming. We summarise some recent scalable GPVAE models in Appendix D.

## 5. Experiments

In this section, we evaluate our models on various tasks involving both synthetic and real-world datasets. We begin by examining their ability to learn latent representations in a moving ball dataset. Next, we apply them to rotated MNIST to impute corrupted frames and generate any missing ones. We further conduct a long-sequence conditional generation experiment using MuJoCo action data, along with imputation tasks on geostatistical datasets, one of which is substantially larger ($N \sim 10^5$) than those typically seen in previous GPVAEs ($N \sim 10$).

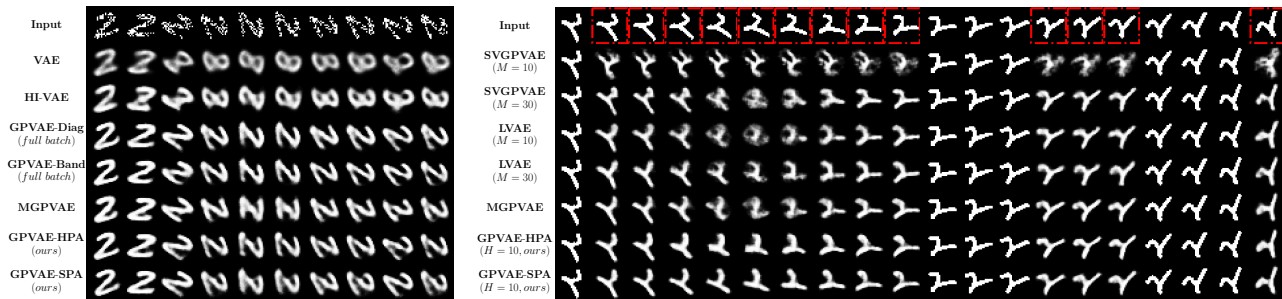

(a) Imputation on series with missing pixels.    (b) Generation on an unseen rotated MNIST sequence with missing frames.

**Figure 3: (a)** Corrupted frame imputation with around 60% missing pixels. **(b)** Missing frames generation on Rotated MNIST with around 60% missing frames. The red boxes indicate the missing frames in that sequence.

We benchmark our models against diverse GPVAEs, mainly focusing on scalable models such as SVGPVAE (Jazbec et al., 2021), LVAE (Ramchandran et al., 2021), MGPVAE (Zhu et al., 2023), and SGPBAE (Tran et al., 2023). Some other baselines, like VAE (Kingma & Welling, 2014), HI-VAE (Nazabal et al., 2020), and NNGP (Wu et al., 2022) are also considered. The performance is assessed using test negative log-likelihood (NLL) and root mean squared error (RMSE). We report the wall-clock training time to estimate computational efficiency. We strive to follow the original baseline settings to ensure the fairest comparisons possible. All experimental results report the mean and standard deviation from 10 random trials. Further experimental setups can be found in Appendix C.

## 5.1. Moving Ball

Our experiments start with latent representation learning on the synthetic moving ball data from Pearce (2020). This dataset consists of black-and-white videos of 30 frames capturing the motion of a pixel ball. Video samples are shown in the leftmost column of Fig. 2a. The two-dimensional trajectory of the ball in each video is simulated from a GP with a radial basis function (RBF) kernel. We aim to infer the ball's trajectory based on the pixel-level frames. In particular, the trajectory is represented by the mean of the latent variable from the encoder.

As each video is relatively short in this dataset and can be grouped into a single batch, conducting full GP inference is feasible. This allows for a direct comparison of our approach with the full-batch baselines, i.e., GPVAE-Casale (Casale et al., 2018) and GPVAE-Pearce (Pearce, 2020).

**Trajectory reconstruction**  Fig. 2a shows the reconstructed latent trajectories of three test videos using 10 nearest neighbours or inducing points. The RMSE appended to each column directly reflects the reconstruction quality. We can see that the VAE fails to learn the latent dynamics behind the videos, as the fully factorised Gaussian prior over

the latent variables cannot account for correlations between samples. The SVGPVAE begins to learn the trend, but the performance is unsatisfactory due to an insufficient number of inducing points. Our two models are quite close to the full-batch baseline, i.e., GPVAE-Casale, with the same number of nearest neighbours.

**Scaling behaviour**  We further show in Fig. 2b the performance of our models compared to two full-GP approaches and SVGPVAE across varying numbers of nearest neighbours and inducing points. While both GPVAE-Pearce and GPVAE-Casale use fully correlated GP priors, they differ in how they construct their variational distributions—GPVAE-Pearce's setup (5) is more closely aligned with SVGPVAE, while GPVAE-Casale adopts a diagonal-encoder approach (3) that matches ours. From Fig. 2b, the SVGPVAE requires nearly half the total number of data points in the trajectory to attain performance levels comparable to the full-batch baseline, whereas our models achieve this with even fewer points. The GPVAE-SPA attains good reconstruction quality using only $\frac{1}{6}$ of the total trajectory points. Table 6 shows additional results using different values of $H$.

## 5.2. Rotated MNIST

Rotated MNIST consists of sequences of handwritten digit images from the MNIST dataset (LeCun et al., 1998), in which each image frame is consecutively rotated at different angles. This classic dataset is designed to assess how well the model can learn latent dynamics and is commonly used in GPVAE literature. Here, we use two versions of the dataset to showcase our models in two distinct scenarios: imputing corrupted frames and generating missing frames.

**Imputing corrupted frames**  This task uses the sequences created by Krishnan et al. (2015). The dataset has 50,000/10,000 training/test sequences, each containing 10 frames. The rotations between two consecutive frames are normally distributed, with around 60% pixels absent randomly. Fig. 3a shows a sequence from the dataset.

**Table 1:** Performance of different models for tasks of the corrupted and missing frames imputation on Rotated MNIST.

| | Models | NLL | RMSE | Training Time (s/epoch) |
|---|---|---|---|---|
| Corrupted Frames | VAE (Kingma & Welling, 2014) | $0.193 \pm 0.038$ | $0.333 \pm 0.011$ | $\mathbf{75.741 \pm 0.660}$ |
| | HI-VAE (Nazabal et al., 2020) | $0.114 \pm 0.002$ | $0.222 \pm 0.002$ | $76.969 \pm 0.594$ |
| | GPVAE-Diag (Fortuin et al., 2020) | $0.094 \pm 0.000$ | $0.199 \pm 0.000$ | $127.614 \pm 2.758$ |
| | GPVAE-Banded (Fortuin et al., 2020) | $0.146 \pm 0.001$ | $\mathbf{0.189 \pm 0.001}$ | $768.347 \pm 24.297$ |
| | MGPVAE (Zhu et al., 2023) | $\mathbf{0.090 \pm 0.001}$ | $0.197 \pm 0.001$ | $110.772 \pm 0.651$ |
| | GPVAE-HPA-H5 (ours) | $0.095 \pm 0.000$ | $0.199 \pm 0.000$ | $137.860 \pm 0.326$ |
| | GPVAE-SPA-H5 (ours) | $0.096 \pm 0.000$ | $0.202 \pm 0.000$ | $121.185 \pm 0.817$ |
| Missing Frames | SVGPVAE-M10 (Jazbec et al., 2021) | $87.711 \pm 0.484$ | $6.122 \pm 0.015$ | $\mathbf{8.280 \pm 0.111}$ |
| | SVGPVAE-M30 (Jazbec et al., 2021) | $77.792 \pm 0.592$ | $5.572 \pm 0.030$ | $9.180 \pm 0.188$ |
| | LVAE-M10 (Ramchandran et al., 2021) | $73.887 \pm 0.748$ | $5.578 \pm 0.022$ | $28.919 \pm 0.253$ |
| | LVAE-M30 (Ramchandran et al., 2021) | $73.545 \pm 1.035$ | $5.558 \pm 0.024$ | $29.063 \pm 0.122$ |
| | MGPVAE (Zhu et al., 2023) | $72.217 \pm 0.192$ | $5.496 \pm 0.005$ | $25.103 \pm 0.103$ |
| | GPVAE-HPA-H10 (ours) | $\mathbf{71.264 \pm 1.263}$ | $5.521 \pm 0.046$ | $9.819 \pm 0.064$ |
| | GPVAE-SPA-H10 (ours) | $\mathbf{69.538 \pm 0.664}$ | $\mathbf{5.412 \pm 0.026}$ | $9.358 \pm 0.083$ |

We follow the settings from Fortuin et al. (2020), where missing pixels are filled with zeros during inference, and the ELBOs are calculated only on the observed pixels in the data. A trade-off parameter is added to the ELBO to rebalance the influence of the likelihood and KL terms (Higgins et al., 2017). For baselines, we additionally consider GPVAE-Diag and GPVAE-Band proposed in Fortuin et al. (2020), both of which are not amenable to mini-batching. GPVAE-Diag uses an encoder with a diagonal covariance matrix, while GPVAE-Band features a tridiagonal covariance. Given that GPVAE-Diag shares the identical structure as our model and employs full batches of sequences, it serves as the gold standard for our models. As suggested by Fortuin et al. (2020), all GP-related models employ Cauchy kernels, except MGPVAE using a Matérn-$\frac{3}{2}$ kernel. We set $H = 5$, which is half the length of a sequence.

Imputation results on the test set are presented in Fig. 3a and Table 1. While VAE and HI-VAE achieve shorter training times, they fail to reconstruct the images faithfully due to a factorised Gaussian prior. In contrast, our models closely match the performance of GPVAE-Diag, demonstrating a strong approximation to the full-batch benchmark. GPVAE-Band achieves the lowest RMSE but incurs the longest training time due to its more complex structure. Although MGPVAE outperforms our model in terms of quantitative metrics, Fig. 3a indicates the generated images do not differ significantly in visual quality, suggesting that the overall performance of our models is comparable to MGPVAE.

**Generating missing frames** Following the setup in Zhu et al. (2023), we produce sequences of MNIST frames, where each digit rotates through two full revolutions. This experiment involves 4,000 training and 1,000 test sequences, each with 100 frames. Each frame is dropped with probabil-

ity 0.6, leaving the remaining sequence with varying lengths. This task aims to generate the missing frames at a given timestamp in unobserved sequences. SVGPVAE/LVAE-M10/30 utilise $M = 10/30$ inducing points, and our models use $H = 10$ nearest neighbours for comparison. We adapt LVAE by manually assigning unique IDs to each new sequence during testing.

Fig. 3b and Table 1 provide the experimental results. As shown in Table 1, both our models surpass the baselines regarding NLL, with GPVAE-SPA achieving the lowest RMSE. This is further validated by Fig. 3b. SVGPVAE-M10 exhibits the fastest training time but the poorest performance. Our models take a slightly longer training time than SVGPVAE-M10, with extra computational overhead caused by additional neighbouring data in a single mini-batch. Although we triple the number of inducing points, which incurs a similar training time to ours, it still cannot outperform ours. LVAEs, which rely on additive kernels over sequence IDs, narrow the performance gap but still produce higher NLLs than our models. MGPVAE experiences longer training times on the long sequences due to its requirement for a sequential forward and backward process, which cannot perform time-step mini-batching in parallel.

### 5.3. MuJoCo Hopper Physics

The MuJoCo dataset collects physical simulation data from the DeepMind Control Suite (Rubanova et al., 2019). In this experiment, 500 series of 14-dimensional Hopper data are generated, each with 1000 timestamps. The series set is then split into training, validation, and test subsets by 320/80/100. At each timestamp, there is a 60% chance that the relevant data features will be completely missing. The task objective is to make posterior predictions at unobserved time steps.

**Table 2:** Results of conditional generation at missing timestamps on MuJoCo dataset.

| Models | NLL | RMSE | Training Time (s/epoch) |
|---|---|---|---|
| SVGPVAE-M10 (Jazbec et al., 2021) | $-1.512 \pm 0.032$ | $0.0583 \pm 0.003$ | $44.000 \pm 5.477$ |
| SVGPVAE-M30 (Jazbec et al., 2021) | $-1.071 \pm 0.175$ | $0.0391 \pm 0.001$ | $47.893 \pm 3.170$ |
| MGPVAE (Zhu et al., 2023) | $-1.708 \pm 0.057$ | $0.0398 \pm 0.002$ | $57.291 \pm 1.141$ |
| GPVAE-HPA-H10 (ours) | $\mathbf{-2.335 \pm 0.032}$ | $\mathbf{0.0222 \pm 0.001}$ | $\mathbf{37.000 \pm 3.793}$ |
| GPVAE-SPA-H10 (ours) | $\mathbf{-1.715 \pm 0.159}$ | $\mathbf{0.0378 \pm 0.007}$ | $\mathbf{39.403 \pm 2.755}$ |

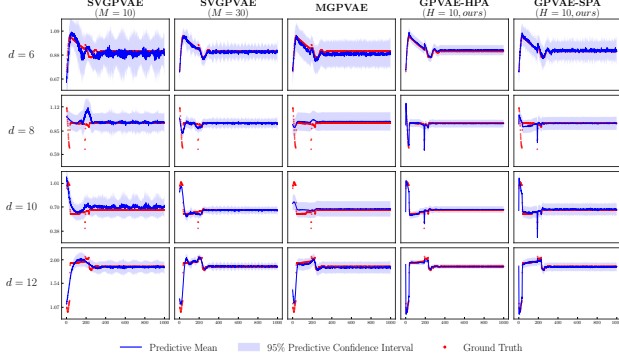

**Figure 4:** Conditional generation at missing timestamps of a MuJoCo sequence in its 6th, 8th, 10th, and 12th dimensions with 95% credible intervals.

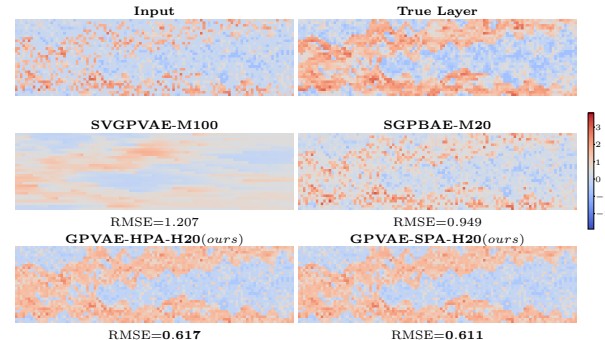

**Figure 5:** Porosity imputation (i.e., the 1st channel) at the intermediate layer of SPE10 dataset using GPVAE models.

**Results** The results in Table 2 demonstrate that our models consistently outperform both SVGPVAE and MGPVAE in terms of NLL, RMSE, and training time. These results are further supported by the visual comparison presented in Fig. 4. Additional numerical results and illustrative plots can be found in Appendix C.3.

SVGPVAE-M10 shows suboptimal RMSE as an insufficient number of inducing points significantly limits the model capacity. Although increasing inducing points enhances the RMSE performance, SVGPVAE-M30 remains inferior to our models using 10 nearest neighbours. Moreover, increasing inducing points in SVGPVAE causes a deterioration in the NLL metric, indicating a reduced capacity to capture predictive uncertainty in this experiment. In contrast, we show in Table 11 in Appendix C.3 that GPVAE-HPA and GPVAE-SPA consistently achieve better NLL and RMSE as the number of nearest neighbours increases. These results highlight the advantage of employing neighbour-based approximation in the latent space over a strategy relying on inducing points. MGPVAE also demonstrates a poorer performance compared to our proposed models and, notably, requires significantly more computational time than ours.

### 5.4. Geostatistical Datasets

We further consider imputation tasks on two multi-dimensional geostatistical datasets of different scales, where

the nearest neighbour principle should be particularly effective. The first dataset, Jura (Goovaerts, 1997), contains hundreds of data points, while the second dataset, SPE10 (Christie & Blunt, 2001), consists of over $10^5$ observations. Additionally, we compare several GP baselines such as exact GP (Rasmussen & Williams, 2006), SVGP (Hensman et al., 2013), and VNNGP (Wu et al., 2022). Appendix C.4 provides more experimental settings and additional results.

**Jura** The Jura dataset provides measurements of three metal concentrations (Nickel, Zinc, and Cadmium) collected in a region of Swiss Jura ($\mathbf{x}_n \in \mathbb{R}^2, \mathbf{y}_n \in \mathbb{R}^3$). The training set includes Nickel and Zinc measurements at all 359 locations, whereas Cadmium measurements are available for only 259 of these sites. We aim to predict Cadmium measurements at the remaining 100 locations.

Table 12 in Appendix C.4.1 compares the performance of our proposed models to competing methods. Our models again surpass the baselines in terms of both RMSE and NLL. The performance of Exact GP and VNNGP is suboptimal due to their inability to capture correlations between output variables. VAE and HI-VAE perform poorly as they do not leverage spatial information. SVGPVAE and SGPBAE exhibit improved performance with an increased number of inducing points; however, they remain inferior to GPVAE-SPA, even using three times the number of inducing points compared to the nearest neighbour count in our models.

**Table 3:** Imputation results of various models on SPE10 dataset.

| Models | NLL | RMSE | Training Time (s/epoch) |
|---|---|---|---|
| MOGP-M1000 (Hensman et al., 2013) | $6.034 \pm 0.003$ | $1.230 \pm 0.000$ | $3.820 \pm 0.021$ |
| VNNGP-H20 (Wu et al., 2022) | $1.052 \pm 0.000$ | $0.683 \pm 0.000$ | $2.490 \pm 0.084$ |
| VAE (Kingma & Welling, 2014) | $3.436 \pm 0.015$ | $1.298 \pm 0.002$ | $\mathbf{1.416 \pm 0.020}$ |
| HI-VAE (Nazabal et al., 2020) | $0.752 \pm 0.034$ | $0.647 \pm 0.008$ | $1.448 \pm 0.011$ |
| SVGPVAE-M100 (Jazbec et al., 2021) | $2.025 \pm 0.020$ | $0.979 \pm 0.006$ | $4.755 \pm 0.097$ |
| SGPBAE-M20 (Tran et al., 2023) | $1.335 \pm 0.106$ | $0.898 \pm 0.125$ | $1585.6 \pm 21.3$ |
| GPVAE-HPA-H20 | $\mathbf{0.735 \pm 0.007}$ | $\mathbf{0.638 \pm 0.007}$ | $4.019 \pm 0.036$ |
| GPVAE-SPA-H20 | $\mathbf{0.736 \pm 0.004}$ | $\mathbf{0.638 \pm 0.004}$ | $3.365 \pm 0.117$ |

**SPE10** SPE10 is a large dataset widely used in petroleum engineering, featuring reservoir properties like porosity and permeability with high *spatial heterogeneity*. We downsample it to 141,900 data points, with $\mathbf{x}_n \in \mathbb{R}^3$ and $\mathbf{y}_n \in \mathbb{R}^4$, masking about 50% of the values for imputation. This data size poses a challenge for GP-based models.

Imputation results are summarised in Table 3. We present plots that illustrate the imputation of porosity at the intermediate layer. In this highly heterogeneous dataset, methods that rely on inducing points, such as Multi-Output GPs (MOGP), SVGPVAE, and SGPBAE, all perform poorly, even using a large number of inducing points (e.g., MOGP-M1000). This is because inducing point methods struggle to effectively manage data that exhibits rapid local changes. In contrast, models employing nearest neighbour approximation (including our models and VNNGP) generally perform better on this dataset, indicating that the nearest neighbour approach helps capture local structure. Importantly, our models outperform VNNGP with $15\times$ fewer trainable parameters, demonstrating that the VAE structure effectively models the latent correlations in the data.

## 6. Conclusion

In this work, we present an efficient neighbour-driven approximation strategy for modelling latent variables in GPVAEs, yielding two new variants: GPVAE-HPA and GPVAE-SPA. By focusing GP interactions on nearest neighbours in the latent space, both models capture essential correlations and facilitate scalable mini-batch training. Empirical results on various tasks ranging from time-series imputation to geostatistical modelling demonstrate consistent accuracy and speed-ups over existing GPVAE baselines.

**Limitations and future work** While our method does not require a specific distance metric, we primarily used Euclidean distance. Exploring alternatives, such as correlation-based distances (Kang & Katzfuss, 2023), and leveraging advanced manifold-aware metrics in high-dimensional data are promising directions for future work.

## Acknowledgements

We thank the reviewers for their insightful comments and constructive suggestions. We also acknowledge the CSF3 at the University of Manchester for providing GPU resources. Xinxing is supported by the UoM-CSC Joint Scholarship, and Xiaoyu receives support through the UoM Departmental Studentship for the Department of Computer Science.

## Impact Statement

This paper presents work aimed at advancing machine learning. There are potential societal consequences of our work, none of which we feel must be specifically highlighted here.

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

## A. Acronyms

We list all acronyms appearing in the paper in the following Table 4 for reference.

**Table 4:** Acronym summary

| Acronym | Meaning |
| --- | --- |
| GP | Gaussian Process |
| VAE | Variational AutoEncoder |
| GPVAE | Gaussian Process Variational AutoEncoder |
| NNGP | Nearest Neighbour Gaussian Process |
| SVGP | Sparse (Stochastic) Variational Gaussian Process |
| SVGPVAE | Sparse Variational Gaussian Process Variational AutoEncoder |
| LVAE | Longitudinal Variational AutoEncoder |
| MGPVAE | Markovian Gaussian Process Variational AutoEncoder |
| SGPBAE | Sparse Gaussian Process Bayesian AutoEncoder |
| HPA | Hierarchical Prior Approximation |
| SPA | Sparse Precision Approximation |
| NLL | Negative Log-Likelihood |
| RMSE | Root Mean Squared Error |

## B. ELBO Derivation

This section details the ELBO derivation of (10) and (13) in Section 3.2. Throughout, we use subscripts to denote the learnable parameters of the distributions: $\phi$ represents the encoder parameters, $\theta$ denotes the decoder parameters, and $\psi$ is associated with the latent GP parameters.

### B.1. GPVAE-HPA ELBO

We begin by outlining the derivation for GPVAE-HPA. In each training iteration, randomly sample a mini-batch $\mathcal{B} = \{(\mathbf{x}_i, \mathbf{y}_i)\}_{i \in \mathcal{I}}$ with the index set $\mathcal{I}$. For each training point $(\mathbf{x}_i, \mathbf{y}_i) \in \mathcal{B}$, look for the $H$ locations $\mathbf{X}_{n(i)} \in \mathbb{R}^{H \times D}$ in $\mathbf{X} = [\mathbf{x}_n]_{n=1}^N$ such that $n(i)$ are indices of top $H$ nearest points to $\mathbf{x}_i$. This neighbour-driven strategy materialises the sampling of a binary indicative vector $\mathbf{w}$ for latent variables. Then, the ELBO is given by

$$
\begin{aligned}
\log p(\mathbf{Y}) &\geq \int q(\mathbf{Z}, \mathbf{w}) \log \frac{p(\mathbf{Y} \mid \mathbf{Z}) p(\mathbf{Z} \mid \mathbf{w}) p(\mathbf{w})}{q(\mathbf{Z}, \mathbf{w})} \, \mathrm{d}\mathbf{Z} \, \mathrm{d}\mathbf{w} \\
&= \int p(\mathbf{w}) \left\{ \int q(\mathbf{Z} \mid \mathbf{w}) \log p(\mathbf{Y} \mid \mathbf{Z}) \, \mathrm{d}\mathbf{Z} + \int q(\mathbf{Z} \mid \mathbf{w}) \log \frac{p(\mathbf{Z} \mid \mathbf{w})}{q(\mathbf{Z} \mid \mathbf{w})} \, \mathrm{d}\mathbf{Z} \right\} \mathrm{d}\mathbf{w} \\
&= \mathbb{E}_{p(\mathbf{w})} \left\{ \mathbb{E}_{q(\mathbf{Z} \mid \mathbf{w})} \log p(\mathbf{Y} \mid \mathbf{Z}) - \mathrm{KL}\left[ q(\mathbf{Z} \mid \mathbf{w}) \,\|\, p(\mathbf{Z} \mid \mathbf{w}) \right] \right\} \\
&= \sum_{i=1}^N \mathbb{E}_{p(\mathbf{w})} \left\{ \mathbb{E}_{q(\mathbf{z}_i \mid \mathbf{w})} \log p(\mathbf{y}_i \mid \mathbf{z}_i) - \frac{1}{N} \mathrm{KL}\left[ q(\mathbf{Z} \mid \mathbf{w}) \,\|\, p(\mathbf{Z} \mid \mathbf{w}) \right] \right\} \\
&\approx \frac{N}{|\mathcal{I}|} \sum_{i \in \mathcal{I}} \left\{ \mathbb{E}_{q(\mathbf{z}_i)} \left[ \log p(\mathbf{y}_i \mid \mathbf{z}_i) \right] - \frac{1}{N} \mathrm{KL}\left[ q(\mathbf{Z}_{n(i)}) \,\|\, p(\mathbf{Z}_{n(i)}) \right] \right\} := \mathcal{L}_{\mathrm{HPA}}.
\end{aligned}
$$

The following part provides a detailed derivation of each term in the above-mentioned ELBO. Specifically, The ELBO $\mathcal{L}_{\mathrm{HPA}}$ using a hierarchical prior is given by

$$
\begin{aligned}
\mathcal{L}_{\mathrm{HPA}} = \frac{N}{|\mathcal{I}|} \sum_{i \in \mathcal{I}} \mathbb{E}_{q_\phi(\mathbf{z}_i \mid \mathbf{y}_i)} \left[ \log p_\theta(\mathbf{y}_i \mid \mathbf{z}_i) \right] && \text{(re-parameterization trick)} \\
- \frac{1}{|\mathcal{I}|} \sum_{i \in \mathcal{I}} \underbrace{\mathrm{KL}\left[ q_\phi(\mathbf{Z}_{n(i)} \mid \mathbf{Y}_{n(i)}) \,\|\, p_\psi(\mathbf{Z}_{n(i)} \mid \mathbf{X}_{n(i)}) \right]}_{\mathrm{KL}[q_\phi \| p_\psi]}. && \text{(closed form)}
\end{aligned}
$$

- *Expected log-likelihood* (if the decoder models Gaussian distributions with mean $\mu_\theta$ and variance $\sigma_\theta^2$ as in the MuJoCo experiment in Section 5.3.)

$$\mathbb{E}_{q_\phi(\mathbf{z}_i|\mathbf{y}_i)}\left[\log p_\theta(\mathbf{y}_i \mid \mathbf{z}_i)\right] = \mathbb{E}_{q_\phi(\mathbf{z}_i|\mathbf{y}_i)}\left[\log \mathcal{N}(\mathbf{y}_i \mid \mu_\theta(\mathbf{z}_i), \sigma_\theta^2(\mathbf{z}_i))\right]$$
$$\approx \sum_{k=1}^{K}\left\{-\frac{\left(\mathbf{y}_i^k - \mu_\theta^k(\mathbf{z}_i)\right)^2}{2\sigma_\theta^{2^k}(\mathbf{z}_i)} - \frac{1}{2}\log\left(2\pi\sigma_\theta^{2^k}(\mathbf{z}_i)\right)\right\}.$$

Here, we slightly abuse notation by writing $\mathbf{z}_i \sim q_\phi(\mathbf{z}_i \mid \mathbf{y}_i)$ to indicate re-parameterized samples. The superscript $k$ represents the $k$-th entry of a vector.

- *KL divergence*

$$\mathrm{KL}\left[q_\phi \mid\mid p_\psi\right] = \sum_{l=1}^{L}\mathrm{KL}\left[\mathcal{N}(\mathbf{z}_{n(i)}^l \mid \boldsymbol{\mu}_q^l, \boldsymbol{\Sigma}_q^l) \mid\mid \mathcal{N}(\mathbf{z}_{n(i)}^l \mid \boldsymbol{\mu}_p^l, \boldsymbol{\Sigma}_p^l)\right]$$
$$= \frac{1}{2}\sum_{l=1}^{L}\left\{(\boldsymbol{\mu}_q^l - \boldsymbol{\mu}_p^l)(\boldsymbol{\Sigma}_p^l)^{-1}(\boldsymbol{\mu}_q^l - \boldsymbol{\mu}_p^l)^\top + \mathrm{tr}\left\{(\boldsymbol{\Sigma}_p^l)^{-1}\boldsymbol{\Sigma}_q^l\right\} + \log|\boldsymbol{\Sigma}_p^l| - \log|\boldsymbol{\Sigma}_q^l| - |n(i)|\right\}$$

where $l$ represents the latent dimension and $\boldsymbol{\Sigma}_q^l$ is diagonal,

$$\begin{cases}\boldsymbol{\mu}_q^l = \mu_\phi^l(\mathbf{Y}_{n(i)}), & \boldsymbol{\Sigma}_q^l = \sigma_\phi^{2\,l}(\mathbf{Y}_{n(i)}); \\ \boldsymbol{\mu}_p^l = \mu_\psi^l(\mathbf{X}_{n(i)}), & \boldsymbol{\Sigma}_p^l = \mathbf{K}_\psi^l(\mathbf{X}_{n(i)}, \mathbf{X}_{n(i)}).\end{cases}$$

## B.2. GPVAE-SPA ELBO

Next, we derive the ELBO for GPVAE-SPA. Following the same notation as in the main text, we have

$$\log p(\mathbf{Y}) \geq \int q(\mathbf{Z})\log\frac{p(\mathbf{Y} \mid \mathbf{Z})\prod_{j=1}^{N}p(\mathbf{z}_j \mid \mathbf{Z}_{n(j)})}{q(\mathbf{Z})}\,\mathrm{d}\mathbf{Z}$$
$$= \int q(\mathbf{Z})\log p(\mathbf{Y} \mid \mathbf{Z})\,\mathrm{d}\mathbf{Z} - \int q(\mathbf{Z})\log\frac{\prod_{j=1}^{N}q(\mathbf{z}_j)}{\prod_{j=1}^{N}p(\mathbf{z}_j \mid \mathbf{Z}_{n(j)})}\,\mathrm{d}\mathbf{Z}$$
$$= \int q(\mathbf{Z})\log p(\mathbf{Y} \mid \mathbf{Z})\,\mathrm{d}\mathbf{Z} - \sum_{j=1}^{N}\int q(\mathbf{Z}_{n(j)})q(\mathbf{z}_j)\log\frac{q(\mathbf{z}_j)}{p(\mathbf{z}_j \mid \mathbf{Z}_{n(j)})}\,\mathrm{d}\mathbf{z}_j\,\mathrm{d}\mathbf{Z}_{n(j)}$$
$$= \sum_{i=1}^{N}\mathbb{E}_{q(\mathbf{z}_i)}\left[\log p(\mathbf{y}_i \mid \mathbf{z}_i)\right] - \sum_{j=1}^{N}\mathbb{E}_{q(\mathbf{Z}_{n(j)})}\mathrm{KL}\left[q(\mathbf{z}_j) \mid\mid p(\mathbf{z}_j \mid \mathbf{Z}_{n(j)})\right].$$

In every training iteration, randomly sample a mini-batch of training data indices $\mathcal{I}$ and a mini-batch of inducing indices $\mathcal{J}$ for stochastically maximising the following ELBO

$$\mathcal{L}_{\mathrm{SPA}} = \frac{N}{|\mathcal{I}|}\sum_{i\in\mathcal{I}}\mathbb{E}_{q_\phi(\mathbf{z}_i|\mathbf{y}_i)}\left[\log p_\theta(\mathbf{y}_i \mid \mathbf{z}_i)\right] - \frac{N}{|\mathcal{J}|}\sum_{j\in\mathcal{J}}\mathbb{E}_{q_\phi(\mathbf{Z}_{n(j)}|\mathbf{Y}_{n(j)})}\left\{\mathrm{KL}[q_\phi(\mathbf{z}_j \mid \mathbf{y}_j) \mid\mid p_\psi(\mathbf{z}_j \mid \mathbf{Z}_{n(j)}, \mathbf{X}_{n(j)})]\right\}.$$

- *Expected log-likelihood* is the same as in $\mathcal{L}_{\mathrm{HPA}}$.

- *KL divergence*

$$\mathbb{E}_{q_\phi(\mathbf{Z}_{n(j)}|\mathbf{Y}_{n(j)})}\left\{\mathrm{KL}[q_\phi(\mathbf{z}_j \mid \mathbf{y}_j) \mid\mid p_\psi(\mathbf{z}_j \mid \mathbf{Z}_{n(j)}, \mathbf{X}_{n(j)})]\right\}$$
$$= \sum_{l=1}^{L}\mathbb{E}_{q_\phi(\mathbf{Z}_{n(j)}^l|\mathbf{Y}_{n(j)})}\left\{\underbrace{\mathrm{KL}[q_\phi(\mathbf{z}_j^l \mid \mathbf{y}_j) \mid\mid p_\psi(\mathbf{z}_j^l \mid \mathbf{Z}_{n(j)}^l, \mathbf{X}_{n(j)})]}_{\mathrm{KL}[q_\phi||p_\psi]}\right\}$$

First, given a data index $j$ and the corresponding neighbour set $\mathbf{Z}_{n(j)}$, we have

$$
\begin{aligned}
\mathrm{KL}\left[q_\phi \,\|\, p_\psi\right] &= \mathrm{KL}\left[\mathcal{N}(\mathbf{z}_j \mid \boldsymbol{\mu}_q, \boldsymbol{\Sigma}_q) \,\|\, \mathcal{N}(\mathbf{z}_j \mid \boldsymbol{\mu}_p, \boldsymbol{\Sigma}_p)\right] \\
&= \frac{1}{2} \sum_{l=1}^{L} \left\{ \frac{(\mu_q^l - \mu_p^l)^2}{\Sigma_p^l} + \frac{\Sigma_q^l}{\Sigma_p^l} + \log|\Sigma_p^l| - \log|\Sigma_q^l| - 1 \right\} \\
&= \frac{1}{2} \left\{ (\boldsymbol{\mu}_q - \boldsymbol{\mu}_p)\boldsymbol{\Sigma}_p^{-1}(\boldsymbol{\mu}_q - \boldsymbol{\mu}_p)^\top + \mathrm{tr}\{\boldsymbol{\Sigma}_p^{-1}\boldsymbol{\Sigma}_q\} + \log\frac{|\boldsymbol{\Sigma}_p|}{|\boldsymbol{\Sigma}_q|} - L \right\},
\end{aligned}
\tag{14}
$$

where $\mathbf{z}_j \in \mathbb{R}^L$, $\mathbf{Z}_{n(j)} \in \mathbb{R}^{|n(j)| \times L}$, and both $\boldsymbol{\Sigma}_p$ and $\boldsymbol{\Sigma}_q$ are diagonal:

$$
\boldsymbol{\mu}_q = \mu_\phi(\mathbf{y}_j), \quad \boldsymbol{\Sigma}_q = \sigma_\phi^2(\mathbf{y}_j);
$$

$$
\begin{cases}
\mu_p^l = m^l(\mathbf{x}_j) + \mathbf{k}_{j,n(j)}^l \mathbf{K}_{n(j),n(j)}^{l\,-1} \left[ \mathbf{Z}_{n(j)}^l - m^l(\mathbf{X}_{n(j)}) \right], \\
\Sigma_p^l = k_{j,j}^l - \mathbf{k}_{j,n(j)}^l \mathbf{K}_{n(j),n(j)}^{l\,-1} \mathbf{k}_{n(j),j}^l, \quad \boldsymbol{\Sigma}_p = \mathrm{diag}(\Sigma_p^l).
\end{cases}
$$

Then, calculate the expectation w.r.t.

$$
\mathbf{Z}_{n(j)} \sim \prod_{l=1}^{L} q_\phi(\mathbf{Z}_{n(j)}^l \mid \mathbf{Y}_{n(j)}) = \prod_{l=1}^{L} \mathcal{N}\left( \mathbf{Z}_{n(j)}^l \mid \boldsymbol{\mu}_{n(j)}^l, \mathbf{S}_{n(j)}^l \right),
$$

where $\boldsymbol{\mu}_{n(j)}^l = \mu_\phi^l(\mathbf{Y}_{n(j)})$ and $\mathbf{S}_{n(j)}^l = \mathrm{diag}(\mathbf{s}_{n(j)}^l) = \sigma_\phi^{2\,l}(\mathbf{Y}_{n(j)})$. Therefore,

$$
\mathbb{E}_{q_\phi(\mathbf{Z}_{n(j)}^l \mid \mathbf{Y}_{n(j)})}\left[ (\mu_p^l - \mu_q^l)^2 \right] = \mathbf{b}_{n(j),j}^{l\,\top} \mathbf{S}_{n(j)}^l \mathbf{b}_{n(j),j}^l + \left\{ m^l(\mathbf{x}_j) + \mathbf{b}_{n(j),j}^{l\,\top}[\boldsymbol{\mu}_{n(j)}^l - m^l(\mathbf{X}_{n(j)})] - \mu_q^l \right\}^2,
$$

where $\mathbf{b}_{n(j),j}^l = \mathbf{K}_{n(j),n(j)}^{l\,-1} \mathbf{k}_{n(j),j}^l$. The above expression can be plugged into (14) to derive the KL term of the ELBO expression.

## C. Experimental Details and Additional Results

Our implementation is open-sourced at `https://github.com/shixinxing/NNGPVAE-official`.

For fair comparisons, most scalable models (including our models, SVGPVAE, MGPVAE, SGPBAE, VAE, HI-VAE and GP models) are implemented in `PyTorch` (Paszke et al., 2019) and `GPyTorch` (Gardner et al., 2018). We use the official code for the baselines GPVAE-Diag and GPVAE-Band from Fortuin et al. (2020) and modify the code to make the running process comparable with others. The experiments are run on an NVIDIA A100-SXM4 or V100-SXM2 GPU of a high-performance cluster. We use the modern similarity search package, Faiss (Johnson et al., 2019), for nearest neighbour searches. Most training time is estimated on an NVIDIA RTX-4090 GPU, except for the missing pixel imputation task, which is tested on an RTX-2080-Ti due to software compatibility.

While the ELBO of GPVAE-SPA can theoretically be computed using two separate mini-batches, in practice, we follow Wu et al. (2022) to simplify the process by using the same mini-batches of data in each iteration. The experimental details are provided in the following sections.

### C.1. Moving Ball

**Experimental settings**  All GP-related models use RBF kernels with the lengthscale initialised to 2. In each epoch, we generate 35 videos using distinct local seeds and train for 25,000 epochs. Our models use the same multilayer perceptron (MLP) structures and training settings as in Jazbec et al. (2021) and Pearce (2020). Two independent GPs control the two-dimensional variable in the latent space of the GPVAE. We summarise the experimental settings in Table 5.

In each training epoch, 35 videos with 30 frames each are simulated for parameter learning. The decoder yields independent Bernoulli distributions for 1024 pixels of each frame. When testing, we derive the latent trajectory by least-squares projection and compute the sums of squared errors on a testing batch of 35 videos. The MSE for one experimental trial is obtained by averaging the squared errors over 10 video batches. The RMSE is then the square root of the MSE.

**Table 5:** Experimental settings for the moving ball experiment.

| Setting | Value |
|---|---|
| Videos in each epoch | 35 |
| Frames in each video | 30 |
| Frame size | $32 \times 32$ |
| Encoder (MLPs) | $1024 \to 500 \to 4$ |
| Latent dimensionality | 2 |
| Decoder (MLPs) | $2 \to 500 \to 1024 \to$ Sigmoid |
| Activation function | tanh |
| Optimizer | Adam, $lr = 0.001$ |
| Training epochs | 25000 |

**Additional results**   We provide RMSEs for the two proposed models in Table 6, with the number of nearest neighbours $H$ ranging from 3 to 15. The results confirm that the reconstruction improves with more nearest neighbours, which enables the models to leverage more information from data in this experiment. We also present the reconstructed latent trajectories in Fig. 6 with 5 or 15 inducing points/nearest neighbours.

**Table 6:** Additional experimental results in terms of RMSE with different numbers of nearest neighbours.

| $H$ | 3 | 5 | 7 | 10 | 15 |
|---|---|---|---|---|---|
| GPVAE-HPA | $17.400 \pm 3.691$ | $8.149 \pm 3.294$ | $4.959 \pm 0.250$ | $2.978 \pm 0.101$ | $\mathbf{1.709 \pm 0.146}$ |
| GPVAE-SPA | $3.507 \pm 0.318$ | $1.905 \pm 0.148$ | $1.289 \pm 0.065$ | $\mathbf{1.079 \pm 0.097}$ | $1.245 \pm 0.063$ |

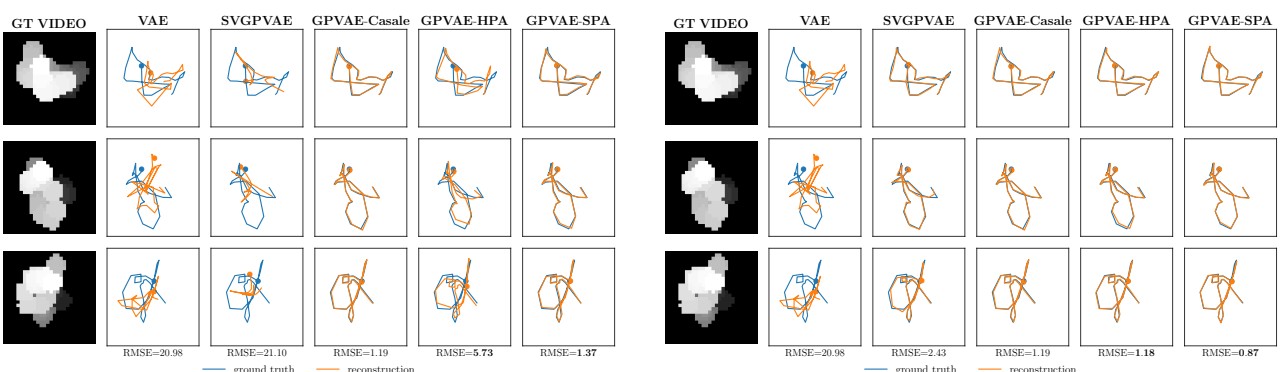

**(a)** Latent trajectory reconstruction with $H = 5$ or $M = 5$.     **(b)** Latent trajectory reconstruction with $H = 15$ or $M = 15$.

**Figure 6:** Additional illustration of latent representation learning for the moving ball series using different numbers of nearest neighbours/inducing points. **(a)** GPVAE-SPA has already achieved good performance using only 5 nearest neighbours. **(b)** When using 15 nearest neighbours/inducing points, all models except VAE achieve good reconstruction quality, but ours still show better RMSE performance.

### C.2. Rotated MNIST

The RMSEs reported in the following experiments are obtained through the encoder mean, computed at missing pixels and averaged. The NLL is evaluated using 20 samples from the latent variables. Specifically,

$$\text{NLL} = -\log \int p(\mathbf{Y} \mid \mathbf{Z}) q(\mathbf{Z}) \, d\mathbf{Z} \approx -\log \frac{1}{S} \sum_s p(\mathbf{Y} \mid \mathbf{Z}_s)$$

$$= \log S - \text{logsumexp}_{s=1,\cdots,S} \log p(\mathbf{Y} \mid \mathbf{Z}_s),$$

where $S$ is the number of samples.

**Table 7:** Experimental settings for the Healing MNIST pixel imputation task.

| Setting | Value |
|---|---|
| Training/test sequences | 50,000/10,000 |
| Frames per sequence | 10 |
| Frame size | $28 \times 28$ |
| Encoder (Convs + MLPs) | Conv(1, 256, ks=3) $\rightarrow$ Conv(256, 1, ks=3) $\rightarrow$ 256 $\rightarrow$ 256 $\rightarrow$ 512 |
| Latent dimensionality | 256 |
| Decoder | $256 \rightarrow 256 \rightarrow 256 \rightarrow 256 \rightarrow 784 \rightarrow$ Sigmoid |
| Activation function | ReLU |
| Optimizer | Adam, $lr = 0.0005$ |
| Mini-batch of sequences | 50 |
| Training epochs | 40 |
| Trade-off parameter $\beta$ | HPA=1.5; SPA=1.0 |

### C.2.1. CORRUPTED FRAME IMPUTATION

The dataset, also called Healing MNIST, simulates real-world situations where medical data is often incomplete (Krishnan et al., 2015). For this task of missing pixel imputation, we follow the settings in Fortuin et al. (2020), where Cauchy kernels and convolutional layers (Conv) are used as components of the model structure. The MGPVAE baseline uses a Matérn-$\frac{3}{2}$ kernel. The lengthscale and outputscale of all kernels are initialised to 2 and 1, respectively. The settings are listed in Table 7. Fig. 7 illustrates additional imputation results for other digits in the dataset.

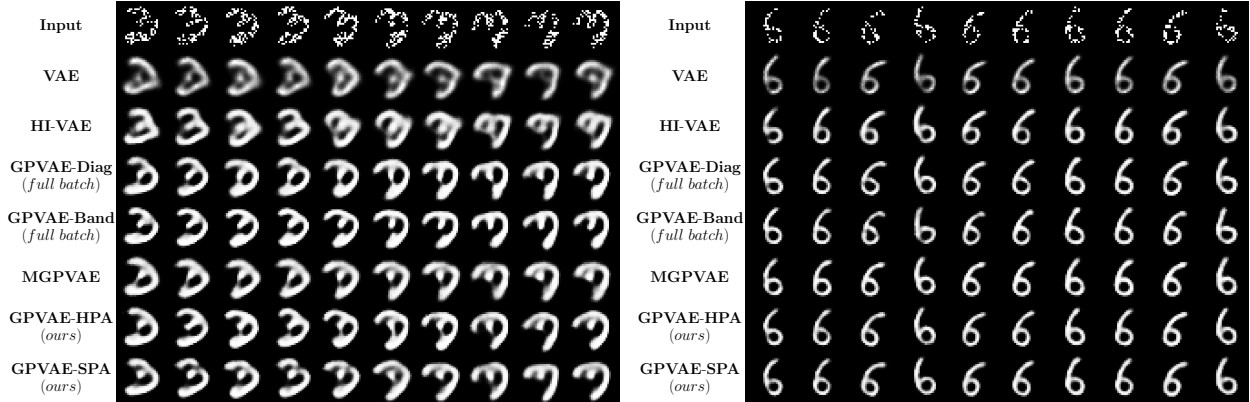

**Figure 7:** Additional imputation results on the Healing MNIST dataset.

### C.2.2. MISSING FRAME GENERATION

**Experimental settings**  We employ a similar architecture to Zhu et al. (2023), with convolutional layers in the encoder and deconvolutional (Deconv) layers in the decoder. We use a Matérn-$\frac{3}{2}$ kernel for MGPVAE and RBF kernels for the others (with the exception of LVAE, which uses an additive kernel incorporating distinct sequence IDs). The lengthscale and outputscale of all kernels are initialised to 2 and 1, respectively. In SVGPVAE, we select the initial locations of the inducing points to be evenly distributed across the entire time range.

**Additional results**  In addition to the $H = 10$ results detailed in the main text, we evaluate our models using varying numbers of nearest neighbours. The metrics for NLL and RMSE are presented in Table 9. This experiment demonstrates that increasing the number of nearest neighbours can enhance the visual quality of the generated images. Fig. 8 illustrates some additional results on the task of generating missing frames based on the timestamps.

**Table 8:** Experimental settings for the missing frame generation experiment on Rotated MNIST.

| Setting | Value |
|---|---|
| Training/test sequences | 4,000/1,000 |
| Frames per sequence | 100 |
| Frame size | $28 \times 28$ |
| Encoder (Convs + MLPs) | Conv(1, 32, ks=3) $\rightarrow$ Conv(32, 64, ks=3) $\rightarrow$ 32 |
| Latent dimensionality | 16 |
| Decoder | $16 \rightarrow 1568 \rightarrow$ Deconv(32, 64, ks=3) $\rightarrow$ Deconv(64, 32, ks=3) $\rightarrow$ Deconv(32, 1, ks=3) $\rightarrow$ Sigmoid |
| Activation function | ReLU |
| Optimizer | Adam, $lr = 0.0005$ |
| Mini-batch of sequences | 50 |
| Mini-batch of frames | 20 |
| Training epochs | 300 |
| Trade-off parameter $\beta$ | HPA=1.5; SPA=1.0 |

**Table 9:** Our models' performance on the task of missing frames generation with different numbers of nearest neighbours.

| | $H$ | 5 | 10 | 20 |
|---|---|---|---|---|
| NLL | GPVAE-HPA | $101.741 \pm 0.712$ | $71.264 \pm 1.263$ | $\mathbf{69.637 \pm 1.027}$ |
| | GPVAE-SPA | $70.865 \pm 0.656$ | $69.538 \pm 0.664$ | $\mathbf{69.516 \pm 0.543}$ |
| RMSE | GPVAE-HPA | $6.669 \pm 0.033$ | $5.521 \pm 0.046$ | $\mathbf{5.444 \pm 0.043}$ |
| | GPVAE-SPA | $5.472 \pm 0.029$ | $\mathbf{5.412 \pm 0.026}$ | $5.425 \pm 0.020$ |

## C.3. MuJoCo Hopper Physics

**Experimental settings** All models use a 15-dimensional latent space and Matérn-$\frac{3}{2}$ kernels. We adopt the same encoder and decoder architectures and training configurations as Zhu et al. (2023). For each latent GP, we initialise the lengthscale and outputscale of the Matérn-$\frac{3}{2}$ kernel to 50 and 1, respectively. All models are trained for 500 epochs. At test time, we draw 20 latent samples to compute the NLL and RMSE. These metrics are computed over the entire sequence for predictive performance on the test set. Additional setups are presented in Table 10.

**Table 10:** Experimental settings for the MuJoCo Hopper Physics experiment.

| Setting | Value |
|---|---|
| Encoder (MLPs) | $14 \rightarrow 32 \rightarrow 30$ |
| Latent dimensionality | 15 |
| Decoder (MLPs) | $15 \rightarrow 16 \rightarrow 14$ |
| Activation function | ReLU |
| Optimizer | Adam, $lr = 0.001$ |
| Mini-batch of sequences | 16 |
| Mini-batch of frames | 32 |
| Training epochs | 500 |

**Table 11:** Additional NLL and RMSE results for varying numbers of nearest neighbours on MuJoCo dataset.

| | $H$ | 5 | 10 | 20 |
|---|---|---|---|---|
| NLL | GPVAE-HPA | $-2.322 \pm 0.061$ | $\mathbf{-2.335 \pm 0.032}$ | $-2.332 \pm 0.021$ |
| | GPVAE-SPA | $-1.700 \pm 0.237$ | $-1.715 \pm 0.159$ | $\mathbf{-1.835 \pm 0.125}$ |
| RMSE | GPVAE-HPA | $0.0225 \pm 0.001$ | $0.0222 \pm 0.001$ | $\mathbf{0.0221 \pm 0.000}$ |
| | GPVAE-SPA | $0.0392 \pm 0.011$ | $0.0378 \pm 0.007$ | $\mathbf{0.0339 \pm 0.005}$ |

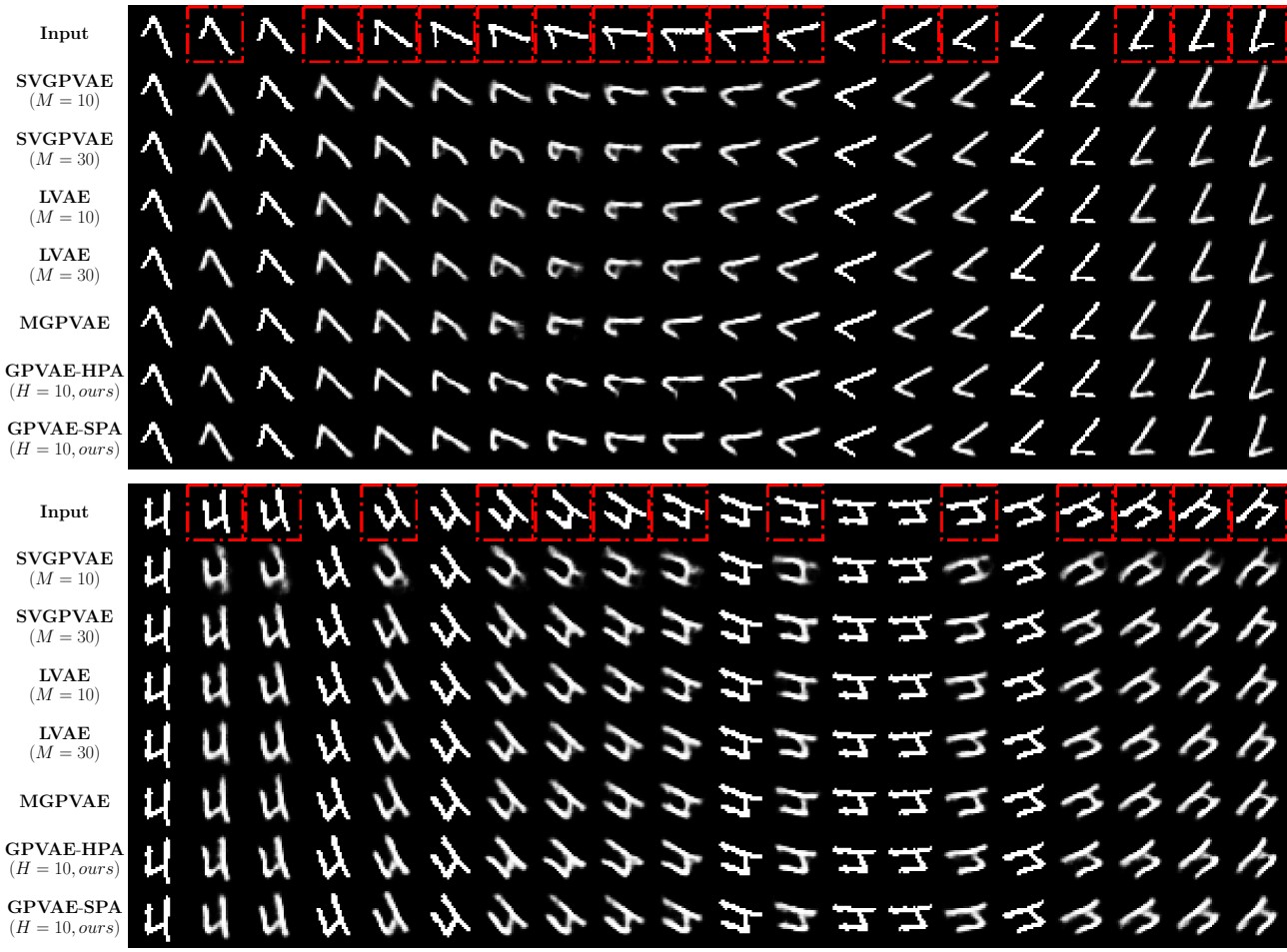

**Figure 8:** Missing frames generated from two unseen MNIST sequences. The red boxes indicate the missing frames.

**Additional results** We provide additional NLL and RMSE results for our two proposed models in Table 11, with $H \in \{5, 10, 20\}$. These results confirm that the predictive performance increases with the number of nearest neighbours, consistent with our early findings. Fig. 11 shows additional plots for a test MuJoCo sequence across its 1st, 2nd, 3rd, 4th, 5th, 7th, 9th, 11th, 13th, and 14th dimensions.

## C.4. Geostatistical Datasets

### C.4.1. JURA

Table 12 presents the results on the Jura dataset discussed in the main text. We also illustrate the prediction in Fig. 9.

**Experimental settings** We follow the experimental setting in (Tran et al., 2023), in which RBF kernels are used, and the lengthscale and outputscale are initialised to 0.1 and 1. More details are in Table 13. SGPBAE (Tran et al., 2023) adopts a stochastic encoder, for which the input size is 6 instead of 3. It uses Adaptive SGHMC (Springenberg et al., 2016) with a learning rate $lr = 0.001$, and the encoder is trained by Adam with a learning rate $lr = 0.001$. After 500 burn-in iterations, 50 samples are collected every 20 iterations. The exact GP and VNNGP are applied to the Cadmium variable only (because they do not model multi-output data natively), while all other models utilise information from all 3 variables.

### C.4.2. SPE10

The SPE10 dataset is derived from the Tenth Comparative Solution Project in petroleum engineering. It is widely used as a benchmark for assessing how different numerical and geostatistical methods perform in heterogeneous reservoir conditions.

**Table 12:** Imputation results of various models on the Jura dataset.

| Models | NLL | RMSE |
|---|---|---|
| Exact GP (Rasmussen & Williams, 2006) | 1.305 | 0.724 |
| VNNGP-H10 (Wu et al., 2022) | $1.387 \pm 0.002$ | $0.700 \pm 0.000$ |
| VAE (Kingma & Welling, 2014) | $2.015 \pm 0.344$ | $0.994 \pm 0.121$ |
| HI-VAE (Nazabal et al., 2020) | $1.082 \pm 0.079$ | $0.710 \pm 0.028$ |
| SVGPVAE-M10 (Jazbec et al., 2021) | $1.032 \pm 0.025$ | $0.678 \pm 0.016$ |
| SVGPVAE-M20 (Jazbec et al., 2021) | $1.002 \pm 0.040$ | $0.661 \pm 0.026$ |
| SVGPVAE-M30 (Jazbec et al., 2021) | $0.981 \pm 0.062$ | $0.650 \pm 0.039$ |
| SGPBAE-M10 (Tran et al., 2023) | $2.171 \pm 0.826$ | $1.089 \pm 0.253$ |
| SGPBAE-M20 (Tran et al., 2023) | $2.005 \pm 0.781$ | $1.061 \pm 0.244$ |
| SGPBAE-M30 (Tran et al., 2023) | $1.435 \pm 0.316$ | $0.913 \pm 0.124$ |
| GPVAE-HPA-H10 | $1.230 \pm 0.184$ | $0.678 \pm 0.054$ |
| GPVAE-SPA-H10 | $\mathbf{0.939 \pm 0.066}$ | $\mathbf{0.584 \pm 0.023}$ |

**Table 13:** Experimental settings for the Jura experiment.

| Setting | Value |
|---|---|
| Encoder (MLPs) | $3 \rightarrow 20 \rightarrow 4$ |
| Latent dimensionality | 2 |
| Decoder (MLPs) | $2 \rightarrow 5 \rightarrow 5 \rightarrow 3$ |
| Activation function | ReLU |
| Optimizer | Adam, $lr = 0.001$ |
| Mini-batch size | 100 (except exact GP, which uses all 259 training data) |
| Training epochs | 2000 for exact GP and VNNGP, 300 for others |
| Trade-off parameter $\beta$ | HPA=1.8, 1.0 for others |

The original dataset contains $220 \times 60 \times 85$ grid cells. Each grid cell represents a discrete unit (a rectangular cuboid) in the 3D reservoir model. Its typical properties are porosity, which indicates the fraction of pore space in the cell, and permeability (often provided in $x$, $y$, and $z$ directions). These properties collectively determine the flow behaviour of fluids (oil, water, or gas) within the reservoir.

**Experimental settings** In this experiment, we downsample the cells by a factor of 2, resulting in $110 \times 30 \times 43$ locations. Each value of the dataset has a 0.5 probability of being dropped. The experimental settings are listed in Table 14.

All models use Cauchy kernels with the lengthscale and outputscale initialised to 2 and 1. The likelihood noise is fixed to 0.25 except for GP models, whose likelihood noise is initialised to 0.01. For SVGPVAE, the inducing locations are uniformly initialised in the data space. For SGPBAE, the SGHMC step size is set to 0.0001, and the steps along the chain are 100. After 1500 burn-in iterations, we keep 100 samples. The performance of SGPVAE is evaluated using 100 samples. Between each sample, the stochastic encoder is updated for 50 iterations.

**Table 14:** Experimental settings for the SPE10 experiment.

| Setting | Value |
|---|---|
| Encoder (MLPs) | $4 \rightarrow 256 \rightarrow 64 \rightarrow 3$ |
| Latent dimensionality | 3 |
| Decoder (MLPs) | $3 \rightarrow 64 \rightarrow 256 \rightarrow 4$ |
| Activation function | ReLU |
| Optimizer | Adam, $lr = 0.001$ |
| Mini-batch size | 1000 |
| Training epochs | 500 |
| Trade-off parameter $\beta$ | HPA=1500, 0.2 for others |

**Additional results** We plot the imputation results of the models at some layers in Fig. 10.

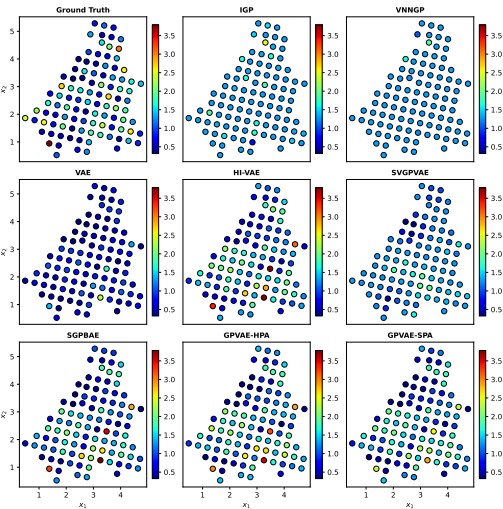

**Figure 9:** Predictive mean for Cadmium on the Jura dataset test split (100 locations).

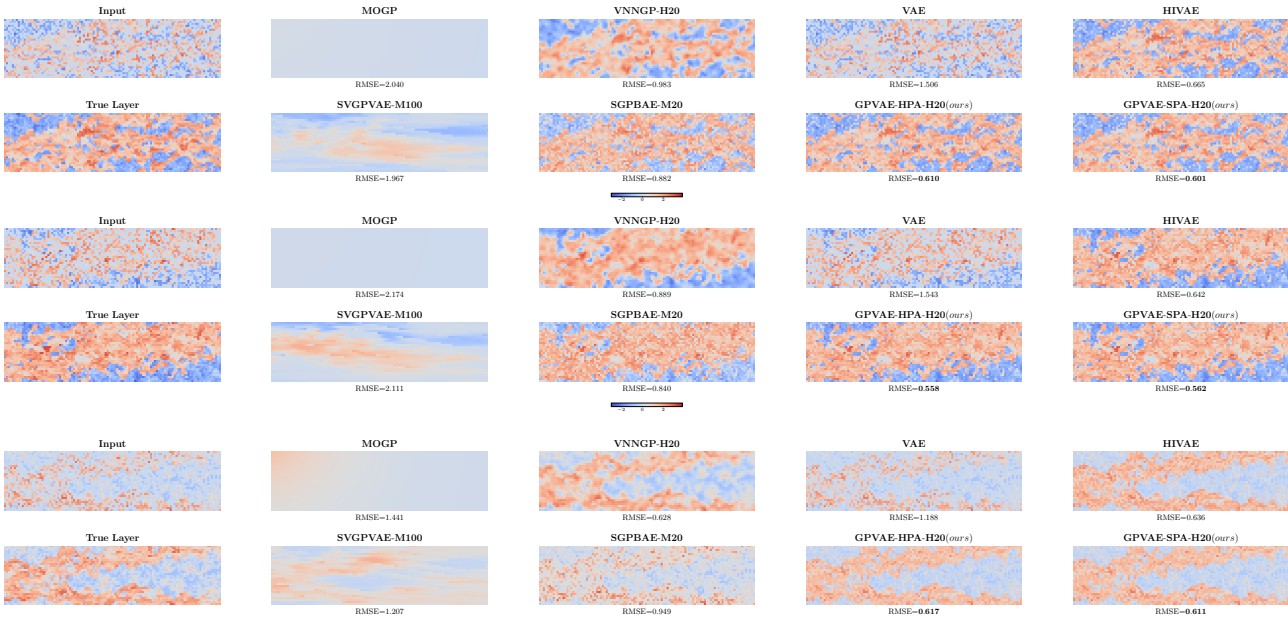

**Figure 10:** Additional imputation results of SPE10. **(top)** The permeability along the $y$ direction at the 26th layer; **(middle)** The permeability along the $x$ direction at the 41st layer; **(bottom)** The porosity at the 22nd layer.

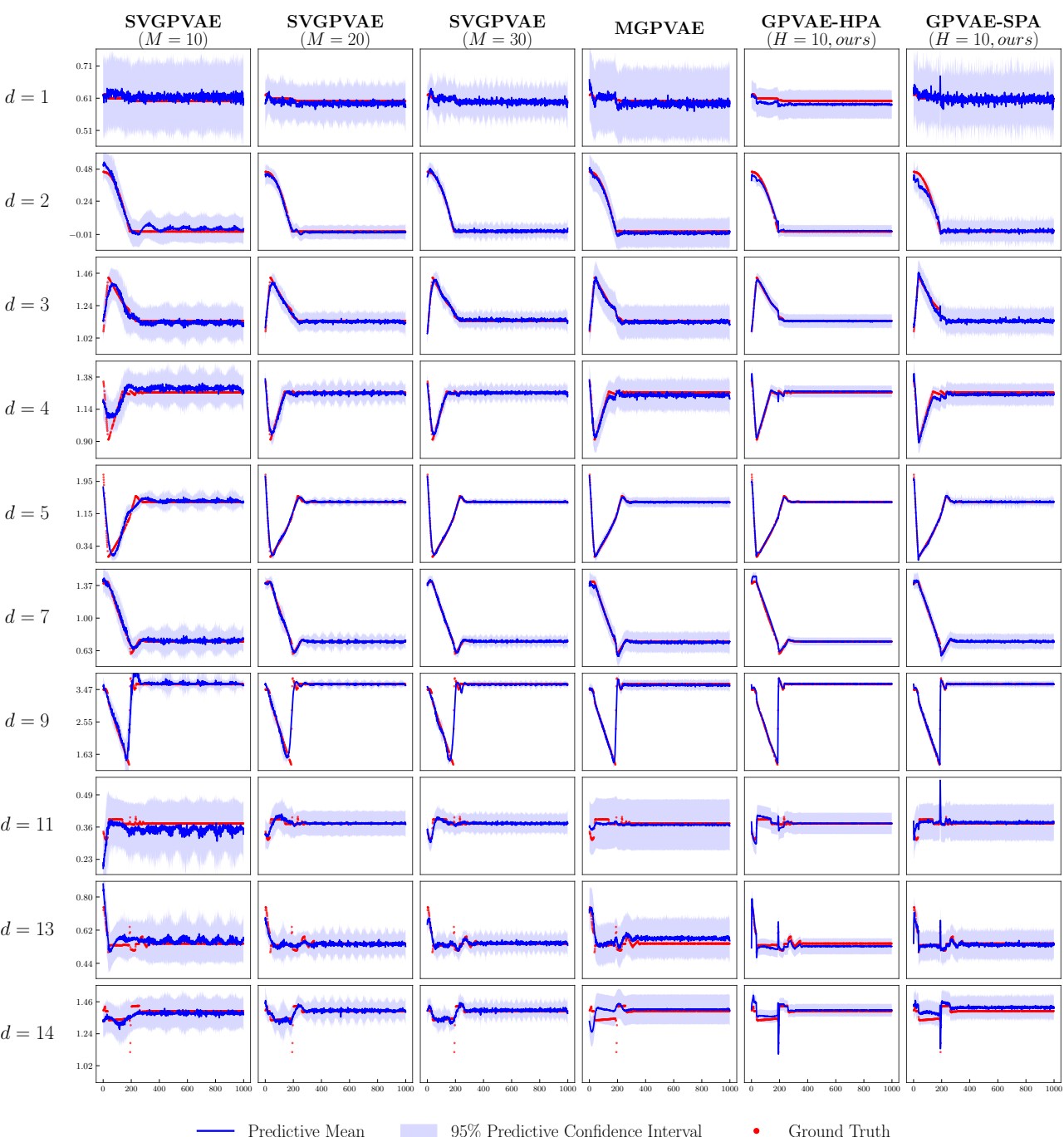

**Figure 11:** 95% posterior credible intervals for a test MuJoCo sequence for all other dimensions with $N = 1000$. The red dots mark unobserved data points.

## D. Scalable Gaussian Process Variational Autoencoders

We give an overview of the related models in the following Table 15.

**Table 15:** A summary of relevant models. $N$, $M$, and $H$ are the number of data points, inducing points, and nearest neighbours. $N_b$ is the mini-batch size. $P$ represents the feature dimensions of low-rank representations. $d$ is the dimensions of the state variables.

| Models | Mini-batching[*] | GP complexity | Arbitrary kernel | Standard VAE encoder | Reference |
|---|---|---|---|---|---|
| VAE | ✓ | / | / | ✓ | Kingma & Welling (2014) |
| GPVAE-Casale | ✓ | $\mathcal{O}(NP^2 + P^3)$ | ✗ | ✓ | Casale et al. (2018) |
| GPVAE-Pearce | ✗ | $\mathcal{O}(N^3)$ | ✓ | ✗ | Pearce (2020) |
| GPVAE-Diag | ✗ | $\mathcal{O}(N^3)$ | ✗ | ✓ | Fortuin et al. (2020) |
| GPVAE-Band | ✗ | $\mathcal{O}(N^3)$ | ✗ | ✗ | Fortuin et al. (2020) |
| SVGPVAE | ✓ | $\mathcal{O}(N_b M^2 + M^3)$ | ✓ | ✗ | Jazbec et al. (2021) |
| MGPVAE | ✗ | $\mathcal{O}(d^3 N)$ | ✗ | ✗ | Zhu et al. (2023) |
| SGPBAE | ✓ | $\mathcal{O}(N_b M^2 + M^3)$ | ✓ | ✗ | Tran et al. (2023) |
| GPVAE-HPA | ✓ | $\mathcal{O}(N_b H^3)$ | ✓ | ✓ | this work |
| GPVAE-SPA | ✓ | $\mathcal{O}(N_b H^3)$ | ✓ | ✓ | this work |

[*] Here means mini-batching along the "time" dimension.

In this section, we summarise recent *scalable* GPVAE models used in our experiments, including SVGPVAE (Jazbec et al., 2021)(Section D.1), MGPVAE (Zhu et al., 2023)(Section D.2), and SGPBAE (Tran et al., 2023)(Section D.3). We believe this section will be helpful for practitioners interested in trying these models. Although SGPBAE is more accurately categorised as an extension of the Bayesian autoencoder (Tran et al., 2021a) rather than a VAE, it is included here for completeness. For more detailed information about these models, please refer to the related papers.

### D.1. SVGPVAE

Replacing exact GPs with Sparse Variational GP (SVGP) techniques (Titsias, 2009; Hensman et al., 2013) to enhance the scalability of GPVAEs is a natural approach. However, such methods cannot be directly integrated into GPVAEs, as they either require the entire dataset to be loaded into memory or are not suited for amortization. In response to this scalability challenge, Jazbec et al. (2021) propose a novel approach that leverages a specific parameterisation of the distribution of inducing variables based on Pearce (2020). This formulation enables the training objective to accommodate mini-batching, amortization, and the use of arbitrary kernels.

In Pearce (2020), they use the variational distribution from (5), and the ELBO is given by:

$$\mathcal{L}_{\text{GPVAE-Pearce}} = \sum_{n=1}^{N} \mathbb{E}_{q_{\psi,\phi}(\mathbf{z}_n | \mathbf{y}_n)} \left[ \log p_\theta(\mathbf{y}_n \mid \mathbf{z}_n) - \sum_{l=1}^{L} \log \tilde{q}_\phi(\mathbf{z}_n^l \mid \mathbf{y}_n) \right] + \sum_{l=1}^{L} \log Z_{\psi,\phi}^l(\mathbf{Y}, \mathbf{X}),$$

where $Z_{\psi,\phi}^l = \mathcal{N}(\mu_\phi^l(\mathbf{Y}) \mid \mathbf{0}, k_\psi^l(\mathbf{X}, \mathbf{X}) + \sigma_\phi^{2l}(\mathbf{Y}))$. The posterior $q_{\psi,\phi}(\mathbf{z}_n \mid \mathbf{y}_n)$ can be regarded as derived from a "pseudo" GP regression with input-observation pairs $(\mathbf{X}, \mu_\phi^l(\mathbf{Y}))$ and noise $\sigma_\phi^{2l}(\mathbf{Y})$. We denote the *latent dataset* as $\{\mathbf{X}, \tilde{\mathbf{Y}} := \mu_\phi(\mathbf{Y}), \tilde{\boldsymbol{\sigma}}^2 := \sigma_\phi^{2l}(\mathbf{Y})\}$. This training objective will require $\mathcal{O}(N^3)$ time complexity due to $\log Z_{\psi,\phi}^l(\mathbf{Y}, \mathbf{X})$ and $q_{\psi,\phi}(\mathbf{z}_n \mid \mathbf{y}_n)$ terms without any approximation. Jazbec et al. (2021) approximate the encoder's output distribution $q_{\psi,\phi}(\mathbf{z})$ using SVGP's posterior:

$$\begin{aligned}
q_s(\mathbf{z}) &= \int p(\mathbf{z} \mid \mathbf{Z_U}) q(\mathbf{Z_U}) \, d\mathbf{Z_U} \\
&= \mathcal{N}(\mathbf{z} \mid \mathbf{K_{xU}} \mathbf{K_{UU}^{-1}} \boldsymbol{\mu}, \mathbf{K_{xx}} + \mathbf{K_{xU}} \mathbf{K_{UU}^{-1}} (\mathbf{A} - \mathbf{K_{UU}}) \mathbf{K_{UU}^{-1}} \mathbf{K_{Ux}}),
\end{aligned}$$
(15)

where $\mathbf{U}$ are $M$ inducing locations and the variational distribution is $q(\mathbf{Z_U}) = \mathcal{N}(\mathbf{Z_U} \mid \boldsymbol{\mu}, \mathbf{A})$. Minimizing $\text{KL}[q_s \mid\mid q_{\psi,\phi}]$

is equivalent to maximizing the (inner) lower bounds $\mathcal{L}_H$ proposed by Hensman et al. (2013)[2]

$$\log Z_{\psi,\phi} \geq \mathcal{L}_H = \sum_{i=1}^{N} \left\{ \log \mathcal{N}\left(\tilde{y}_i \mid \mathbf{K}_{\mathbf{x}_i\mathbf{U}}\mathbf{K}_{\mathbf{UU}}^{-1}\boldsymbol{\mu}, \tilde{\sigma}_i^2\right) - \frac{1}{2\tilde{\sigma}_i^2}\left[\tilde{k}_{ii} + \mathrm{tr}(\mathbf{A}\boldsymbol{\Lambda}_i)\right] \right\} - \mathrm{KL}[q(\mathbf{Z}_\mathbf{U}) \,\|\, p_\psi(\mathbf{Z}_\mathbf{U})], \quad (16)$$

where $\tilde{y}_i$ and $\tilde{\sigma}_i$ are from the latent dataset, $\tilde{k}_{ii} = \mathbf{K}_{\mathbf{x}_i\mathbf{x}_i} - \mathbf{K}_{\mathbf{x}_i\mathbf{U}}\mathbf{K}_{\mathbf{UU}}^{-1}\mathbf{K}_{\mathbf{Ux}_i}$ and $\boldsymbol{\Lambda}_i = \mathbf{K}_{\mathbf{UU}}^{-1}\mathbf{K}_{\mathbf{Ux}_i}\mathbf{K}_{\mathbf{x}_i\mathbf{U}}\mathbf{K}_{\mathbf{UU}}^{-1}$. The practical training objective for SVGPVAE is given by replacing $q_{\psi,\phi}$ with $q_s$ from (15) and lowering bound $\log Z_{\psi,\phi}^l$ with $\mathcal{L}_H$ from (16) simultaneously:

$$\mathrm{ELBO} \gtrsim \sum_{n=1}^{N} \mathbb{E}_{q_s(\mathbf{z}_n)}\left[\log p_\theta(\mathbf{y}_n \mid \mathbf{z}_n) - \sum_{l=1}^{L}\log \tilde{q}_\phi(\mathbf{z}_n^l \mid \mathbf{y}_n)\right] + \sum_{l=1}^{L}\mathcal{L}_H^l. \quad (17)$$

**Mini-batching in ELBO** Take a mini-batch of size $N_b$, $\mathbf{X}_b \subset \mathbf{X}$, $\mathbf{Y}_b \subset \mathbf{Y}$, and the encoder $\tilde{q}_\phi$ creates a mini-batch of the latent dataset $\{\mathbf{X}_b, \tilde{\mathbf{Y}}_b, \tilde{\boldsymbol{\sigma}}_b\}$. SVGPVAE computes stochastic estimates for $\boldsymbol{\mu}, \mathbf{A}$ of the variational distribution $q(\mathbf{Z}_\mathbf{U}) = \mathcal{N}(\mathbf{Z}_\mathbf{U} \mid \boldsymbol{\mu}, \mathbf{A})$ at each latent channel $l$:

$$\begin{aligned}
\boldsymbol{\Sigma}_b^l &:= \mathbf{K}_{\mathbf{UU}} + \frac{N}{N_b}\mathbf{K}_{\mathbf{UX}_b}\,\mathrm{diag}(\tilde{\boldsymbol{\sigma}}_b^{-2})\mathbf{K}_{\mathbf{X}_b\mathbf{U}}, \\
\boldsymbol{\mu}_b^l &:= \frac{N}{N_b}\mathbf{K}_{\mathbf{UU}}(\boldsymbol{\Sigma}_b^l)^{-1}\mathbf{K}_{\mathbf{UX}_b}\,\mathrm{diag}(\tilde{\boldsymbol{\sigma}}_b^{-2})\tilde{\mathbf{Y}}_b^l, \\
\mathbf{A}_b^l &:= \mathbf{K}_{\mathbf{UU}}(\boldsymbol{\Sigma}_b^l)^{-1}\mathbf{K}_{\mathbf{UU}}.
\end{aligned} \quad (18)$$

These estimators converge to the true values for $N_b \to N$. $\boldsymbol{\Sigma}_b^l$ is an unbiased estimator for $\boldsymbol{\Sigma}^l$ but this doesn't hold for $\boldsymbol{\mu}_b^l$ and $\mathbf{A}_b^l$. Then, the training objective of SVGPVAE for a mini-batch $\mathbf{X}_b, \mathbf{Y}_b$ based on the above Monte Carlo estimators is given by

$$\mathcal{L}_{\mathrm{SVGPVAE}} := \frac{N}{N_b}\sum_{n=1}^{N_b}\mathbb{E}_{q_s(\mathbf{z}_n)}\left[\log p_\theta(\mathbf{y}_n \mid \mathbf{z}_n) - \sum_{l=1}^{L}\log \tilde{q}_\phi(\mathbf{z}_n^l \mid \mathbf{y}_n)\right] + \sum_{l=1}^{L}\mathcal{L}_H^l.$$

**Prediction** After training, generating $\mathbf{y}_*$ for an unseen $\mathbf{x}_*$ follows the procedure:

- Encode (possibly all) training sample to get $\{\mu_\phi^l(\mathbf{Y})\}_{l=1}^L$, $\{\sigma_\phi^{2\,l}(\mathbf{Y})\}_{l=1}^L$.

- Compute $\boldsymbol{\Sigma}_N^l \in \mathbb{R}^{M\times M}$, $\boldsymbol{\mu}_N^l \in \mathbb{R}^M$ and $\mathbf{A}_N^l \in \mathbb{R}^{M\times M}$ using the estimators (18) for $l \in \{1, 2, ..., L\}$.

- Compute posterior predictive distribution:

$$q(\mathbf{z}_* \mid \mathbf{Y}, \mathbf{X}, \mathbf{x}_*) = \int p(\mathbf{z}_* \mid \mathbf{Z}_\mathbf{U})\mathcal{N}(\mathbf{Z}_\mathbf{U} \mid \boldsymbol{\mu}_N, \mathbf{A}_N)\,\mathrm{d}\mathbf{Z}_\mathbf{U},$$

  which is Gaussian with mean and variance

$$\mu_*^l = \mathbf{K}_{\mathbf{x}_*\mathbf{U}}(\boldsymbol{\Sigma}_N^l)^{-1}\mathbf{K}_{\mathbf{UX}}\,\mathrm{diag}(\sigma_\phi^{-2}(\mathbf{Y}))\mu_\phi^l(\mathbf{Y}),$$

$$\sigma_*^{2\,l} = \mathbf{K}_{\mathbf{x}_*\mathbf{x}_*} - \mathbf{K}_{\mathbf{x}_*\mathbf{U}}\mathbf{K}_{\mathbf{UU}}^{-1}\left(\mathbf{K}_{\mathbf{UU}} - \mathbf{K}_{\mathbf{UU}}(\boldsymbol{\Sigma}_N^l)^{-1}\mathbf{K}_{\mathbf{UU}}\right)\mathbf{K}_{\mathbf{UU}}^{-1}\mathbf{K}_{\mathbf{Ux}_*}.$$

- Compute $\mathbf{z}_* = \mu_* + \sigma_* \cdot \epsilon$, with $\epsilon$ sampled from a standard normal distribution, and generate $\mathbf{y}_* \sim p_\theta(\mathbf{y}_* \mid \mathbf{z}_*)$.

### D.2. MGPVAE

This work is inspired by the fact that some one-dimensional GPs with Matérn kernels can be written as linear Stochastic Differential Equations (SDEs) (Särkkä & Solin, 2019), which has an equivalent discrete linear state space representation. For Gaussian likelihood, the linear discrete-time representation enables Kalman filtering and smoothing that computes the posterior distributions in linear time w.r.t. series length. This work places such discrete GP representations on the latent space of GPVAE. However, the GPVAE decoder is a non-linear function of latent variables. They further apply site-based approximation (Chang et al., 2020) for the non-linear likelihood terms to enable analytic solutions for the filtering and smoothing procedures.

---

[2]We suppress the channel index $l$ here for simplicity.

### D.2.1. SDE REPRESENTATIONS FOR MARKOVIAN GPs

A GP $z \sim \mathcal{GP}(0, k)$ with a Markovian kernel $k$ (e.g., Matérn-$\frac{3}{2}$) can be written with an Itô SDE of latent dimension $d$ (Solin et al., 2016; Hamelijnck et al., 2021) :

$$d\mathbf{s}_t = \mathbf{F}\mathbf{s}_t dt + \mathbf{L}d\mathbf{B}_t, \quad z_t = \mathbf{H}\mathbf{s}_t, \tag{19}$$

where $\mathbf{F} \in \mathbb{R}^{d \times d}$ is the feedback matrix, $\mathbf{L} \in \mathbb{R}^{d \times e}$ the noise effect, $\mathbf{H} \in \mathbb{R}^{1 \times d}$ the emission matrix. $\mathbf{B}_t$ is an $e$-dimensional (correlated) Brownian motion with spectral density matrix $\mathbf{Q}_c \in \mathbb{R}^{e \times e}$ such that $\mathbf{B}_{t+\delta} - \mathbf{B}_t \sim \mathcal{N}(\mathbf{0}, \delta \mathbf{Q}_c)$.

Suppose the initial state $\mathbf{s}_{t_0} \sim \mathcal{N}(\mathbf{m}_0, \mathbf{P}_0)$ with the stationary state mean and covariance. Given all timestamps $\{t_0, \cdots, t_T\}$, if we solve for each $t_i$ with initial time $t_{i-1}$, then the the linear SDE (19) admits:

$$\mathbf{s}_{t_i} = e^{(t_i - t_{i-1})\mathbf{F}} \mathbf{s}_{t_{i-1}} + \int_{t_{i-1}}^{t_i} e^{(t_i - \tau)\mathbf{F}} \mathbf{L}\, d\mathbf{B}_\tau.$$

Therefore, we can derive a corresponding discrete-time solution, allowing the recursive updates of $\mathbf{s}_{t_{i+1}}$ given $\mathbf{s}_{t_i}$ (Särkkä et al., 2006):

$$\mathbf{s}_{t_{i+1}} = \mathbf{A}_{i,i+1} \mathbf{s}_{t_i} + \mathbf{q}_i, \quad \mathbf{s}_{t_0} \sim \mathcal{N}(\mathbf{m}_0, \mathbf{P}_0),$$
$$\mathbf{A}_{i,i+1} = e^{\Delta_i \mathbf{F}}, \quad \mathbf{q}_i \sim \mathcal{N}(\mathbf{0}, \mathbf{Q}_{i,i+1}), \tag{20}$$

where $\Delta_i = t_{i+1} - t_i$. Provided that $\mathbf{P}_0$ exists and is known, $\mathbf{Q}_{i,i+1}$ can be easily obtained in closed form (Solin et al., 2016):

$$\mathbf{Q}_{i,i+1} = \mathbf{P}_0 - \mathbf{A}_{i,i+1} \mathbf{P}_0 \mathbf{A}_{i,i+1}^\top.$$

Note that $d, \mathbf{F}, \mathbf{L}, \mathbf{H}, \mathbf{m}_0, \mathbf{P}_0, \mathbf{Q}_c$ in (19) and (20) depends on the kernel's type and parameters.

### D.2.2. MARKOVIAN GPVAEs

MGPVAEs utilize the linear SDE form of Markovian GPs in the latent space and transform it into a discrete State-space model (SSM) with non-conjugate measurements (i.e., non-linear likelihood modelled by a decoder). Specifically, given $N$ (possibly unevenly spaced) time steps $\{t_1, t_2, ..., t_N\}$, for the $l$-th latent dimension,

$$\mathbf{s}_{i+1}^l = \mathbf{A}_{i,i+1}^l \mathbf{s}_i^l + \mathbf{q}_i^l,$$
$$\mathbf{s}_0^l \sim \mathcal{N}(\mathbf{m}_0^l, \mathbf{P}_0^l), \quad \mathbf{q}_i^l \sim \mathcal{N}(\mathbf{0}, \mathbf{Q}_{i,i+1}^l),$$
$$z_i^l = \mathbf{H}^l \mathbf{s}_i^l, \quad \mathbf{y}_i \mid \mathbf{z}_i \sim p_\theta(\mathbf{y}_i \mid \mathbf{z}_i),$$

where we abuse the timestamp subscripts, using $\mathbf{s}_i \in \mathbb{R}^{dL}$ to denote the state variable at time $t_i$. $\mathbf{z}_i \in \mathbb{R}^L$ is the latent variable of the GPVAE, and $\theta$ represents the parameters of the decoder. The $L$ latent Markovian GPs are independent, and we will suppress the channel $l$ in subsequent expressions to keep the notation uncluttered.

**Model**   Provided a SSM with system states $\{\mathbf{s}_i\}_{i=1}^N$ and observations $\{\mathbf{Y}_i\}_{i=1}^N$, the forward process is described by the joint distribution

$$p(\{\mathbf{s}_i\}_{i=1}^N, \{\mathbf{Y}_i\}_{i=1}^N) = \prod_{i=1}^N p(\mathbf{s}_i \mid \mathbf{s}_{i-1}) \prod_{i=1}^N p_\theta(\mathbf{Y}_i \mid \mathbf{s}_i).$$

For $i = 0, \ldots, N-1$, the factors of the above distribution are defined by

$$p(\mathbf{s}_{i+1} \mid \mathbf{s}_i) = \mathcal{N}(\mathbf{s}_{i+1} \mid \mathbf{A}_{i,i+1}\mathbf{s}_i, \mathbf{Q}_{i,i+1}),$$
$$p(\mathbf{y}_i \mid \mathbf{s}_i) = p_\theta(\mathbf{Y}_i \mid \mathbf{H}\mathbf{s}_i).$$

Again, the transition matrices $\{\mathbf{A}_{i,i+1}, \mathbf{Q}_{i,i+1}\}$ and the emission matrix $\mathbf{H}$ are determined by the Markovian kernel . The measurement $p(\mathbf{y}_i \mid \mathbf{s}_i)$ is modeled by a non-linear *decoder* network, so the smoothing distribution $p(\mathbf{s}_i \mid \mathbf{y}_{1:N})$ is no longer analytical. Like Pearce (2020), this work utilises pseudo observations $\tilde{\mathbf{y}}_{1:N}$ and pseudo noise variances $\tilde{\mathbf{v}}_{1:N}$ from the output of the encoder to substitute the measurement model $p(\mathbf{y}_i \mid \mathbf{s}_i)$ with

$$p(\tilde{\mathbf{y}}_i \mid \mathbf{s}_i) = \mathcal{N}(\tilde{\mathbf{y}}_i \mid \mathbf{H}\mathbf{s}_i, \tilde{\mathbf{v}}_i)$$

such that the linear Gaussian relationship is maintained and the standard Kalman filter and the RTS smoother can be applied. The smoothing distribution conditioned on the pseudo-observations

$$p(\mathbf{s}_{1:N} \mid \tilde{\mathbf{y}}_{1:N}) = \frac{p(\mathbf{s}_{1:N}) \prod_{i=1}^{N} \mathcal{N}(\tilde{\mathbf{y}}_i \mid \mathbf{H}\mathbf{s}_i, \tilde{\mathbf{v}}_i)}{\int p(\mathbf{s}_{1:N}) \prod_{i=1}^{N} \mathcal{N}(\tilde{\mathbf{y}}_i \mid \mathbf{H}\mathbf{s}_i, \tilde{\mathbf{v}}_i) d\mathbf{s}_{1:N}} = q(\mathbf{s}_{1:N}),$$

will be used as a variational approximation $q(\mathbf{s}_{1:T})$ to the true posterior $p(\mathbf{s}_{1:N} \mid \mathbf{y}_{1:N})$ later.

**Variational Inference**   This work employs variational inference to approximate the intractable posteriors. The variational distribution $q(\mathbf{s}_{1:N})$ is proposed to approximate the posterior $p(\mathbf{s}_{1:N} \mid \mathbf{y}_{1:N})$ over $\mathbf{s}_{1:N} \in \mathbb{R}^{N \times dL}$. Note that $\mathbf{z}_i$ is a deterministic linear transformation of $\mathbf{s}_i$ so the stochasticity arises from $\mathbf{s}_{1:N}$. The variational $q(\mathbf{s}_{1:N})$ is learned by maximizing the following ELBO:

$$
\begin{aligned}
\mathcal{L}_{\text{MGPVAE}} &= \sum_{i=1}^{N} \mathbb{E}_{q(\mathbf{s}_i)} \log p_\theta(\mathbf{y}_i \mid \mathbf{s}_i) - \text{KL}\left[q(\mathbf{s}_{1:N}) \,\|\, p(\mathbf{s}_{1:N})\right] \\
&= \sum_{i=1}^{N} \mathbb{E}_{q(\mathbf{s}_i)} \log p_\theta(\mathbf{y}_i \mid \mathbf{s}_i) - \mathbb{E}_{q(\mathbf{s}_{1:N})}\left[\log \prod_{i=1}^{N} \mathcal{N}(\tilde{\mathbf{y}}_i \mid \mathbf{H}\mathbf{s}_i, \tilde{\mathbf{v}}_i) - \log \int p(\mathbf{s}_{1:N}) \prod_{i=1}^{N} \mathcal{N}(\tilde{\mathbf{y}}_i \mid \mathbf{H}\mathbf{s}_i, \tilde{\mathbf{v}}_i) d\mathbf{s}_{1:N}\right] \\
&= \sum_{i=1}^{N} \mathbb{E}_{q(\mathbf{s}_i)} \log p_\theta(\mathbf{y}_i \mid \mathbf{s}_i) - \sum_{i=1}^{N} \mathbb{E}_{q(\mathbf{s}_i)} \log \mathcal{N}(\tilde{\mathbf{y}}_i \mid \mathbf{H}\mathbf{s}_i, \tilde{\mathbf{v}}_i) + \log \int p(\mathbf{s}_{1:N}) \prod_{i=1}^{N} \mathcal{N}(\tilde{\mathbf{y}}_i \mid \mathbf{H}\mathbf{s}_i, \tilde{\mathbf{v}}_i) \, d\mathbf{s}_{1:N} \\
&= \sum_{i=1}^{N} \mathbb{E}_{q(\mathbf{s}_i)} \left[\log p_\theta(\mathbf{y}_i \mid \mathbf{s}_i) - \log \mathcal{N}(\tilde{\mathbf{y}}_i \mid \mathbf{H}\mathbf{s}_i, \tilde{\mathbf{v}}_i)\right] + \log p(\tilde{\mathbf{y}}_{1:N}).
\end{aligned}
$$

**Prediction**   Computing the state-space posterior distribution at a newly introduced time point $t_*$ relies on the smoothed distributions obtained at the adjacent time points. More concretely, let $t_-$ and $t_+$ denote the immediate predecessor and successor time points, respectively, and the predictive $q(\mathbf{s}_{t_*})$ can be obtained as $q(\mathbf{s}_{t_*}) = \int p(\mathbf{s}_{t_*} \mid \mathbf{s}_{t_-}, \mathbf{s}_{t_+}) q(\mathbf{s}_{t_-}, \mathbf{s}_{t_+}) d\mathbf{s}_{t_-} d\mathbf{s}_{t_+}$. More details can be found in Appendix A.1 of Adam et al. (2020). Using the relationship $\mathbf{z}_{t_*} = \mathbf{H}\mathbf{s}_{t_*}$, we can obtain the predictive posterior $q(\mathbf{z}_{t_*})$ and $q(\mathbf{y}_*)$.

### D.3. SGPBAE

This work presents a fully Bayesian autoencoder (BAE) that treats the GP parameters, the latent variables, and the decoder parameters in a Bayesian fashion (i.e., SGPBAE). To carry out scalable inference, the authors (1) use the fully independent training conditionals (FITC) (Quinonero-Candela & Rasmussen, 2005), (2) adopt stochastic Hamiltonian Monte Carlo (SGHMC) (Chen et al., 2014), and (3) employ an amortized stochastic network as the encoder to learn to draw samples from the posterior of the latent variable. Additionally, this encoder avoids making strong assumptions about the form of the posterior by producing samples from the (approximate) implicit posterior. However, MCMC usually requires a sufficient number of iterations to reach the stationary distribution (convergence), and determining whether it has converged can be challenging. Besides, manual tuning of many parameters adds complexity to the process.

**GP Prior**   The prior over the latent variables $\mathbf{Z} = \left[\mathbf{z}^l\right]_{l=1}^{L} \in \mathbb{R}^{N \times L}$ in this paper is from $L$ independent GPs $\mathbf{F} = \left[\mathbf{f}^l\right]_{l=1}^{L} \in \mathbb{R}^{N \times L}$ equipped with $M$ inducing points and additive Gaussian noise. Specifically, the variables at the $l$-th channel have the following joint distribution:

$$p_\psi(\mathbf{z}^l, \mathbf{f}^l, \mathbf{u}^l \mid \mathbf{S}, \psi) = p_\psi(\mathbf{z}^l \mid \mathbf{f}^l, \psi) p_\psi(\mathbf{f}^l, \mathbf{u}^l \mid \mathbf{S}, \psi),$$

$$p_\psi(\mathbf{f}^l, \mathbf{u}^l \mid \mathbf{S}, \psi) = \mathcal{N}\left(\left[\begin{array}{c}\mathbf{f}^l \\ \mathbf{u}^l\end{array}\right] \middle| \mathbf{0}, \left[\begin{array}{cc}\mathbf{K}_{\mathbf{XX}|\psi} & \mathbf{K}_{\mathbf{XS}|\psi} \\ \mathbf{K}_{\mathbf{SX}|\psi} & \mathbf{K}_{\mathbf{SS}|\psi}\end{array}\right]\right), \; p_\psi(\mathbf{z}^l \mid \mathbf{f}^l, \psi) = \mathcal{N}(\mathbf{z}^l \mid \mathbf{f}^l, \sigma_\psi^2 \mathbf{I}),$$

where $\mathbf{U} = \left[\mathbf{u}^l\right]_{l=1}^{L} \in \mathbb{R}^{M \times L}$ and $\mathbf{S} \in \mathbb{R}^{M \times D}$ are inducing variables and locations. $\psi$ contains the kernel parameters (e.g., lengthscale, variance) and the noise variance $\sigma_\psi^2$. A fully Bayesian treatment in this work leads to a lognormal prior $p_\gamma(\psi)$ over $\psi$ and a uniform prior $p_\xi(\mathbf{S})$ over $\mathbf{S}$.

***Independent Conditionals*** The overall joint distribution should be decomposed over observations to sample from the posterior over all the latent variables using SGHMC. The authors further impose independence in the conditional distribution (Quinonero-Candela & Rasmussen, 2005):

$$p_\psi(\mathbf{f}^l \mid \mathbf{u}^l, \mathbf{S}, \psi) \approx \mathcal{N}\left(\mathbf{f}^l \mid \mathbf{K}_{\mathbf{XS}|\psi}\mathbf{K}_{\mathbf{SS}|\psi}^{-1}\mathbf{u}^l, \mathrm{diag}\left[\mathbf{K}_{\mathbf{XX}|\psi} - \mathbf{K}_{\mathbf{XS}|\psi}\mathbf{K}_{\mathbf{SS}|\psi}^{-1}\mathbf{K}_{\mathbf{SX}|\psi}\right]\right).$$

**Decoder** is a neural network $p(\mathbf{Y} \mid \mathbf{Z}, \boldsymbol{\theta})$ with weights/biases $\boldsymbol{\theta}$ regarded as random variables [3]. Therefore, defining $\boldsymbol{\Psi} = \{\psi, \mathbf{S}, \mathbf{U}, \boldsymbol{\theta}\}$, the *potential energy* function is given by

$$
\begin{aligned}
U(\boldsymbol{\Psi}, \mathbf{Z}) &= -\log p(\psi, \mathbf{S}, \mathbf{U}, \boldsymbol{\theta}) - \sum_{l=1}^{L} \log \int p(\mathbf{z}^l \mid \mathbf{f}^l, \sigma_\psi^2) p(\mathbf{f}^l \mid \mathbf{u}^l, \mathbf{S}, \psi)\, \mathrm{d}\mathbf{f}^l - \log p(\mathbf{Y} \mid \mathbf{Z}, \boldsymbol{\theta}) \\
&= -\left[\log p_\gamma(\psi) + \log p_\xi(\mathbf{S}) + \log p_\psi(\mathbf{U} \mid \psi, \mathbf{S}) + \log p(\boldsymbol{\theta})\right] \\
&\quad - \sum_{n=1}^{N}\left\{\sum_{l=1}^{L} \log \mathcal{N}\left(z_n^l \mid \mathbf{K}_{\mathbf{x}_n\mathbf{S}}\mathbf{K}_{\mathbf{SS}}^{-1}\mathbf{u}^l, k_{\mathbf{x}_n} - \mathbf{K}_{\mathbf{x}_n\mathbf{S}}\mathbf{K}_{\mathbf{SS}}^{-1}\mathbf{K}_{\mathbf{Sx}_n} + \sigma_\psi^2\right) + \log p(\mathbf{y}_n \mid \mathbf{z}_n, \boldsymbol{\theta})\right\} \\
&= -\log p(\boldsymbol{\Psi}) - \frac{N}{|\mathcal{B}|}\sum_{n\in\mathcal{B}}\left\{\sum_{l=1}^{L} \log \mathcal{N}\left(z_n^l \mid \mathbf{K}_{\mathbf{x}_n\mathbf{S}}\mathbf{K}_{\mathbf{SS}}^{-1}\mathbf{u}^l, k_{\mathbf{x}_n} - \mathbf{K}_{\mathbf{x}_n\mathbf{S}}\mathbf{K}_{\mathbf{SS}}^{-1}\mathbf{K}_{\mathbf{Sx}_n} + \sigma_\psi^2\right) + \log p(\mathbf{y}_n \mid \mathbf{z}_n, \boldsymbol{\theta})\right\} \\
&\approx \tilde{U}(\boldsymbol{\Psi}, \mathbf{Z}),
\end{aligned}
$$

where $\tilde{U}(\boldsymbol{\Psi}, \mathbf{Z})$ is a stochastic estimate from a mini-batch. In practice, $\mathbf{u}^l$ is sampled by whitening the prior, i.e., $\mathbf{u}^l = \mathbf{L}\mathbf{v}, \mathbf{L}\mathbf{L}^\top = \mathbf{K}_{\mathbf{SS}}, \mathbf{v} \sim \mathcal{N}(\mathbf{0}, \mathbf{I})$ (Hensman et al., 2015).

**Encoder** is an implicit stochastic network concatenating random noise and $\mathbf{Y}$. If the encoder is a multilayer perceptron (MLP), the authors concatenate the random seeds and the input into a long vector. The dimension of the random seeds is the same as that of the input. If the encoder is a CNN, they spatially stack the random seeds and the input.

**Sampling by SGHMC** SGHMC (Chen et al., 2014) uses a noisy but unbiased estimation of the gradient $\nabla \tilde{U}(\boldsymbol{\Theta})$ computed from a mini-batch of the data, where we group the sampled variables into $\boldsymbol{\Theta} = \{\boldsymbol{\Psi}, \mathbf{Z}\}$. Introducing auxiliary momentum variables $\mathbf{r}$, the discretized Hamiltonian dynamics are then updated as follows:

$$
\begin{cases}
\Delta\boldsymbol{\Theta} &= \eta\mathbf{M}^{-1}\mathbf{r}, \\
\Delta\mathbf{r} &= -\eta\nabla\tilde{U}(\boldsymbol{\Theta}) - \eta\mathbf{C}\mathbf{M}^{-1}\mathbf{r} + \mathcal{N}(\mathbf{0}, 2\eta(\mathbf{C} - \tilde{\mathbf{B}})),
\end{cases}
$$

where $\eta$ is the step size, $\mathbf{M}$ an arbitrary mass matrix that serves as a precondition, $\mathbf{C}$ a user-defined friction matrix, and $\tilde{\mathbf{B}}$ the estimate for the noise of the gradient evaluation. This work adopts an adaptive version of SGHMC (Springenberg et al., 2016), where these hyperparameters are automatically adjusted during a burn-in phase. After this period, these hyperparameters are fixed. Specifically, the updates of the hyperparameters are:

$$
\begin{cases}
\mathbf{M}^{-1} &= \mathrm{diag}\left(\hat{V}_{\boldsymbol{\Theta}}^{-\frac{1}{2}}\right), \\
\Delta\hat{V}_{\boldsymbol{\Theta}} &= -\tau^{-1}\hat{V}_{\boldsymbol{\Theta}} + \tau^{-1}\left[\nabla\tilde{U}(\boldsymbol{\Theta})\right]^2, \\
\Delta\tau &= -g_{\boldsymbol{\Theta}}^2\hat{V}_{\boldsymbol{\Theta}}^{-1}\tau + 1, \\
\Delta g_{\boldsymbol{\Theta}} &= -\tau^{-1}g_{\boldsymbol{\Theta}} + \tau^{-1}\nabla\tilde{U}(\boldsymbol{\Theta}); \\
\tilde{\mathbf{B}} &= \frac{1}{2}\eta\hat{V}_{\boldsymbol{\Theta}}; \\
\eta\mathbf{C}\hat{V}_{\boldsymbol{\Theta}}^{-\frac{1}{2}} &= \alpha\mathbf{I}.
\end{cases}
$$

Here, $\alpha$ is a momentum coefficient. By substituting $\mathbf{v} := \eta\hat{V}_{\boldsymbol{\Theta}}^{-\frac{1}{2}}\mathbf{r}$, the discretized Hamiltonian dynamics becomes

$$
\begin{cases}
\Delta\boldsymbol{\Theta} &= \mathbf{v}, \\
\Delta\mathbf{v} &= -\eta^2\hat{V}_{\boldsymbol{\Theta}}^{-\frac{1}{2}}\nabla\tilde{U}(\boldsymbol{\Theta}) - \alpha\mathbf{v} + \mathcal{N}(\mathbf{0}, 2\eta^2\alpha\hat{V}_{\boldsymbol{\Theta}}^{-\frac{1}{2}} - \eta^4\mathbf{I}).
\end{cases}
$$

---

[3]To remain notation consistency. The original paper uses $\varphi$ to represent the decoder's parameters and $\boldsymbol{\theta}$ for kernel parameters instead.

