# OpenReview forum: "Neighbour-Driven Gaussian Process Variational Autoencoders for Scalable Structured Latent Modelling"
_ICML.cc/2025/Conference — ICML 2025 poster_

### Official Review · Reviewer_Van9 · 2025-03-04

**Overall Recommendation:** 4

**Summary:**

The authors propose a new approximation for Gaussian Process Variational Autoencoders to improve computational efficiency. A GP-VAE is a VAE for the structured data case (e.g. where we have both images Y, and auxiliary information X, such as time, or location) that replaces the independent Gaussian prior over the latents with a GP prior (with the kernel operating over X, giving functions from e.g. time or location, to latents), better utilising correlations between datapoints. The authors use the nearest-neighbour kernel approximation from the GP literature, which only keeps the covariances from the set of H nearest neighbours for each points, leading to a block-sparse kernel matrix, which has similar computational complexity to an inducing-point GP. They propose and evaluate two different ways of combining nearest-neighbour GPs with the GP-VAE framework, and find that their model outperforms other approximate GP-VAE models with similar time complexity.

### Update after rebuttals
I have maintained my score of "accept" - please see rebuttal comment.

**Claims And Evidence:**

- The authors' method indeed appears to improve task performance compared to other approximate GP-VAE methods.
- For comparisons against non-GP VAE methods, the authors at one point state "VAE and HI-VAE perform poorly as they do not leverage spatial information"; it is not clear to me whether this means these methods were not provided X at all, which would obviously result in worse performance.

**Essential References Not Discussed:**

The references provided seem suitable.

**Experimental Designs Or Analyses:**

- The experiments are sensible, and cover a good variety of datasets.
- The authors compare against many other baseline methods, which is good. It is not always clear why some baselines are only used in certain experiments but not others; for example, in Table 1, the baselines are different between the "Corrupted Frames" experiment and the "Missing Frames" experiment, and I'm not sure why (perhaps I missed an explanation in the paper).
- All experiments report means and standard deviations over 10 random trials, including the baselines (which they ran themselves), which is excellent. Additionally, they also report the wall-clock time of all the experiments, which is important given that the point of the approximation is computational efficiency.

**Methods And Evaluation Criteria:**

- The proposed solution of using nearest-neighbour GPs in a GP-VAE is very sensible, given the existing literature on nearest-neighbour GPs, and makes sense for the proposed VAE task. Given that inducing point GPs have already been tried in this domain, this paper fills in an obvious gap in the literature.
- The authors test their method on a good range of tasks, all of which fall under the "structured latent modelling" umbrella.
- The two proposed variants look quite similar at a high level, but sometimes result in moderately different performance. It would have been useful to explain why two different variants were proposed, compare and contrast the two methods in the paper after both have been presented. In particular, there are subtle differences between the ELBOs of the two methods that would be worth highlighting.

**Other Comments Or Suggestions:**

- The table comparing time complexities in Appendix D (Table 15) is very nice and would be useful to have somewhere in the main text, if space allows.
- Section 2.2 (in the background section) talks about inducing point GPVAEs, but as I understood it, we were not using this framework in this paper. If this is not necessary background to understand this paper's methods, it may be best to remove it to avoid adding irrelevant and confusing details.

**Other Strengths And Weaknesses:**

### Strengths
- The paper is very well written. I particularly liked the fact that the authors made it very clear how different models related to each other, with sentences such as "In contrast, setting H = 0 will cause the model to degenerate into VAEs."

### Weaknesses
- The proposed methods are not always that much better than other baselines. For example, in Table 3, the HI-VAE baseline performs as well as their method, despite not utilising the spatial information.

**Questions For Authors:**

- Why are the baselines different between the two experiments in Table 1?
- How are you performing bolding in the tables? In Table 3, you have bolded HI-VAE's training time, though this is not the lowest training time on that table.
- Do the non-GP VAE baselines have access to X at all, e.g. as arguments to the encoder / decoder?

**Relation To Broader Scientific Literature:**

- Nearest neighbour GPs were already proposed in earlier papers
- GP-VAEs were already proposed in earlier papers
- The novelty in this paper stems from combining nearest-neighbour GPs with GP-VAEs. Though this does not seem to have required a huge amount of novel theoretical work, it is an obvious gap in the literature, and this is a well written and executed attempt to fill this gap.

**Theoretical Claims:**

- I did not verify the authors' derivations for their method, which were consigned to the appendix. However, the resulting formulae looked sensible, and did seem to support the time-complexities stated by the authors.

---

> ### Author Rebuttal · Authors · 2025-03-31
>
> We genuinely appreciate your insightful and encouraging feedback and highlighting aspects of our paper that might have gone unnoticed. Below, we hope to address the points in the review.
>
> ---
> ### 1. Using auxiliary information $X$
>
> We acknowledge that our non-GP baselines, VAE and HI-VAE, do *not* use the auxiliary information $X$ - yet they are widely adopted as baselines in the GPVAE literature (e.g., [1,2,3,4]) to underscore the benefits of imposing a GP prior.
>
> Actually, as demonstrated by several GPVAE works (e.g., GPVAE-Casale [6],  SVGPVAE [2], SGPBAE [1]), providing low-dimensional $X$ directly as input to a standard VAE (e.g., in a *conditional VAE* fashion [5]) rarely outperforms GPVAE models on the similar tasks we explored in the paper. As you suggested, we also implemented a Conditional VAE (CVAE) baseline for our *Missing Frames* task, following the design in [6]. Specifically, we processed the scaled timestamp $t$ into $[\sin(t), \cos(t)]$ and used an MLP to produce an additional time feature, then stacked it to the vector after the encoder and before the decoder. The missing frames were generated based on average latent representations and new time features. Below are the results comparing CVAE to our models from 10 random trials:
> | Model/Metric| NLL| RMSE| Training Time(s/epoch)|
> |-|-|-|-|
> |CVAE|213.891$\pm$13.121|10.276$\pm$0.189| **4.736$\pm$0.033**|
> |GPVAE-HPA-H10(ours)|**71.264$\pm$1.263**|**5.521$\pm$0.046**|9.819$\pm$0.064|
> |GPVAE-SPA-H10(ours)|**69.538$\pm$0.664**|**5.412$\pm$0.026**|9.358$\pm$0.083|
>
> As shown, although CVAE trains faster, it performs significantly worse, aligning with previous findings that *directly* injecting low-dimensional $X$ into a standard VAE generally underperforms GPVAE. These results illustrate that the GP prior provides a more effective mechanism to correlate latent variables based on $X$.
>
> ---
> ### 2. Clarify baseline differences across experiments
>
> We omitted certain baselines for practical reasons:
>
> **VAE / HI-VAE in Missing Frames**
>
> In the *Corrupted Frames* setting, partially corrupted images can still be passed to the encoder at test time. However, the *Missing Frames* experiment requires generating entire frames from the latent structure with the new timestamp $x_*$. Since VAE and HI-VAE do not incorporate $X$, they cannot directly generate fully missing frames. Hence, we excluded them in the *Missing Frames* baselines.
>
> **GPVAE-Diag / GPVAE-Banded for Longer Sequences**
>
> Both GPVAE-Diag and GPVAE-Banded [3] were designed mainly for relatively *short* sequences. In our trials with 100-frame sequences, these models showed prohibitively long training times. Since the *Missing Frames* experiment was specifically designed to focus on scalable models suitable for longer and irregular sequences, we chose to exclude methods that do not scale in practice. In future revisions, we will clarify these reasons in the paper.
>
> ---
> ### 3. Typographical issue, code release, and paper structure
>
> We appreciate you noting the boldface inconsistency in Table 3—this was indeed a formatting mistake, and we will correct it. Additionally, as stated in our appendix, we plan to release all source code for our experiments, including our re-implementations of numerous baselines in PyTorch for fairness and ease of comparison. We hope this effort could facilitate further research on GPVAE-based methods.
>
> We agree that Table 15 in Appendix D is quite informative. We will try to include it in the main paper (subject to page limits). We also acknowledge your comments about Section 2.2 on inducing points. Although we included it for context (many large-scale GP methods rely on inducing points), we will streamline or move some details to the appendix to keep our main text more focused on neighbour-driven approaches.
>
> ---
> ### 4. Differences between HPA and SPA
>
> Concretely, HPA (Hierarchical Prior Approximation) imposes sparsity in the *covariance* matrix by “switching off” non-neighbouring latent variables through a hierarchical selection variable. This yields a sparse *covariance* structure. SPA (Sparse Precision Approximation) factorises the GP distribution into chained conditional terms, effectively resulting in a *sparse precision* matrix rather than a sparse covariance matrix. In practice, both are valid neighbour-driven approaches that reduce computational cost. We will further clarify these differences in the final manuscript.
>
> ---
> We hope these clarifications address the points you raised.
>
> [1] Fully Bayesian Autoencoders with Latent Sparse Gaussian Processes (ICML 2023)
>
> [2] Scalable Gaussian Process Variational Autoencoders (AISTATS 2021)
>
> [3] GP-VAE: Deep Probabilistic Time Series Imputation (AISTATS 2020)
>
> [4] The Gaussian Process Prior VAE for Interpretable Latent Dynamics from Pixels (AABI 2020)
>
> [5] Learning Structured Output Representation using Deep Conditional Generative Models (NeurIPS 2015)
>
> [6] Gaussian Process Prior Variational Autoencoders (NeurIPS 2018)

---

> > ### Comment · Reviewer_Van9 · 2025-04-02
> >
> > Thank you for your response. You have thoroughly addressed my concerns, and after briefly considering the other reviews and responses, I'd like to maintain my recommendation as "Accept" and emphasise to the AC that I believe this work has been carried out to a high standard.

---

> > > ### Author Response · Authors · 2025-04-02
> > >
> > > Thank you very much for your encouraging follow-up. We are truly grateful for the time and effort you dedicated to reviewing our submission, and we will incorporate your suggestions in the final version.

---

### Official Review · Reviewer_B5r9 · 2025-03-09

**Overall Recommendation:** 3

**Summary:**

This paper successfully reduces the computational cost of Gaussian Process Variational Autoencoder (GPVAE) by incorporating the Nearest Neighbour Gaussian Process (NNGP). The experimental results demonstrate that the proposed method achieves high performance while reducing computational costs, compared to existing approaches aimed at making GPVAE more scalable.

## update after rebuttal
While I will keep my score as Weak Accept, I would like to note that my overall opinion leans toward acceptance of this paper.

**Claims And Evidence:**

The claim that "existing GPVAEs are computationally expensive" is clear and well-supported by evidence. The proposed method, which addresses this issue using NNGP, is reasonable and well-motivated.

**Essential References Not Discussed:**

I believe the necessary references are cited appropriately.

**Experimental Designs Or Analyses:**

The paper presents extensive experiments on multiple datasets, with various discussions. I have a few concerns, which I will outline in the Questions Section.

**Methods And Evaluation Criteria:**

The paper evaluates the proposed method using various datasets and evaluation metrics.

**Other Comments Or Suggestions:**

Including a pseudo-code for the proposed method would make the paper easier to understand.

**Other Strengths And Weaknesses:**

The idea of using NNGP to reduce the computational cost of GPVAE is simple yet effective.

**Questions For Authors:**

The use of NNGP to reduce the computational cost of GPVAE is a simple yet effective approach, and it appears to achieve high performance experimentally. I am generally in favor of accepting this paper, but I have the following concerns.

1. If I understand correctly, the nearest neighbors are selected in the data space. However, since the Gaussian Process is applied to the latent variables, I find this slightly counterintuitive. Why did you choose to find neighbors in the data space rather than the latent space? Also, is the distance in the data space preserved in the latent variable space, especially in cases where the latent space exhibits structural transformations? (For example, two points that are far apart in the data space might be close in the latent space, and vice versa.)

- If computational cost were not a concern, I would expect that GPVAE would achieve the best performance. However, in Figure 2, the proposed method slightly outperforms GPVAE in reconstruction error, and in Table 1, the NLL follows the order: GPVAE-Diag < Proposed Method < GPVAE-Banded. What is the reason for this?

- While the proposed method achieves consistently high performance, some results are worth discussing. In Table 1 (Corrupted Frames), MGPVAE achieves better performance and lower computational cost than the proposed method, whereas in Missing Frames, the proposed method outperforms others. In Table 3, the proposed method slightly outperforms HI-VAE in performance, but HI-VAE is faster in training time. Given this situation, to what extent can hyperparameter tuning (i.e., increasing or decreasing the number of neighbors) improve the performance?

- Since the number of neighbors is a hyperparameter that trades off between performance and computational cost, do you have any guidelines for setting this value?

I would appreciate your response.

**Relation To Broader Scientific Literature:**

The references provided appear to be sufficient.

**Theoretical Claims:**

I have reviewed the theoretical aspects of the paper, and they appear to be valid. However, I have one concern, which I will outline in the Questions Section.

---

> ### Author Rebuttal · Authors · 2025-03-31
>
> We sincerely appreciate your thoughtful feedback. We hope to address your concerns below.
>
> ---
> ### 1. Selecting neighbours in the data space vs. the latent space
>
> **A natural extension from NNGPs with local correlation**
>
> Our approach follows the principle of NNGPs, which typically rely on adjacency in the *input coordinates* $X$ (spatial, temporal, etc.) for large-scale GPs. Extending this principle to GPVAE, we assume that points close in $X$ are likely to have correlated latent representations under the GP prior, which generally holds well in practice. The GP prior *encourages* points that are near in $X$ to stay close in the latent space. This regularisation enforces correlation in the latent variables such that points with neighbouring inputs are unlikely to end up arbitrarily far apart in the latent space.
>
> **Possible extensions with complex kernels**
>
> We recognise that data points far in data space can sometimes be mapped near in latent space, especially when the encoder is strongly non-linear. In those cases, one might extend the kernel to be non-stationary or incorporate periodic terms. This more flexible kernel design can learn that certain points—though far in raw input coordinates—are effectively neighbours according to the kernel values. In principle, one could then pick neighbours by kernel similarity rather than Euclidean distance. For the tasks we explored, standard kernels and data-space adjacency already perform well.
>
> In summary, using data-space neighbours offers a straightforward, well-tested approach inspired by NNGPs.
>
> ---
> ### 2. Explaining performance gains over full GPVAE in some cases
>
> **Proposed method vs. GPVAE-Diag**
>
> A *full* GP prior can be a stronger regulariser, but it also complicates optimisation, making converging to a good solution harder. By focusing on local correlations rather than forcing a single large covariance across all data points, our method “loosens” the constraints on $Z$ and may achieve better results in some tasks.
>
> **Proposed method vs. GPVAE-Banded**
>
> Both GPVAE-Diag [1] and our model share a standard encoder leading to a *diagonal* posterior (Eq(3)). By contrast, GPVAE-Banded [1] employs a *tridiagonal* encoder covariance, which introduces extra parameters (+~50%) to capture between-sample correlations more explicitly. Although this may yield slightly better performance on *short* sequences, it does not scale easily to longer sequences. Our design focuses on a diagonal encoder with a neighbour-based GP prior, achieving a more balanced trade-off between accuracy and computational feasibility.
>
> ---
> ### 3. Further discussion on some experimental results & hyperparameter guidelines
>
> **Short corrupted vs. long missing frames: different experimental settings**
>
> - The *Corrupted Frames* experiment (following [1]) features a large dataset (60,000 sequences) but *short* sequences (10 frames each). MGPVAE handles short sequences relatively efficiently because its sequential computations do not accumulate large overhead when the sequence length is small. Meanwhile, since the image transformations are relatively smooth from one frame to the next (no missing frames), MGPVAE’s Markov structure enjoys an advantage in capturing local frame-to-frame correlations.
> - By contrast, the *Missing Frames* experiment (following [2]) uses *longer* sequences (100 frames each), and each frame has a 60% chance of being dropped. This creates irregular, sometimes large gaps between observed frames, which presents two challenges for MGPVAE: (1) Given a longer sequence, the forward–backward process needs more runtime to propagate information; (2) The model may accumulate more error at large time gaps where many frames are missing in uneven patterns. Our neighbour-driven method does *not* rely on strictly sequential processes and can have faster training and better predictive performance in this setting.
> - HI-VAE can train faster as it does *not* learn inter-sample dependencies through a GP prior, but this may come at the cost of accuracy.
>
> **Guidelines for the number of neighbours $H$**
>
> The number of nearest-neighbours $H$ directly influences how local or global the GP prior is. In practice, picking $H$ in the same (or smaller) range as the number of inducing points used by sparse-GP baselines is a good starting point: smaller $H$ reduces runtime but may weaken correlation modelling, while larger $H$ may improve accuracy at the expense of more computation. We recommend beginning with a moderate $H$ based on (1) estimated correlation length scales and (2) available computational budget, then adjusting based on validation performance.
>
> ---
> ### 4. Request for Pseudo-Code
>
> We completely agree that providing pseudo-code would help readers reproduce our approach. We will include a succinct algorithm pseudo-code in the appendix of the final version.
>
> [1] GP-VAE: Deep Probabilistic Time Series Imputation (AISTATS 2020)
>
> [2] Markovian Gaussian Process Variational Autoencoders (ICML 2023)

---

> > ### Comment · Reviewer_B5r9 · 2025-04-01
> >
> > Thank you for your response.
> > My concerns have been resolved.
> > While I will keep my score as Weak Accept,
> > I would like to note that my overall opinion leans toward acceptance of this paper.

---

> > > ### Author Response · Authors · 2025-04-02
> > >
> > > Thank you for letting us know that our clarifications have addressed your concerns. We appreciate your leaning toward acceptance, and we will incorporate your feedback in the final version. Thank you again for your time and insight.

---

### Official Review · Reviewer_d16V · 2025-03-11

**Overall Recommendation:** 4

**Summary:**

Variational Autoencoders are deep generative models the learn low dimensional latent representations or high dimensional data, e.g. images in pixel space. When we have extra auxiliarly information, such as images from video and we also know the timestamps of the frames, instead of inferring independant latent representations for each image, we can use the timestamps to force structure of of the series of latent representations and the Gaussian Process Prior VAE is a class of model that uses the auxiliary infomration for inform a correlated GP prior over the corresponding latent representations of the high dimensional datapoints.

However GPs have $O(N^3)$ computational cost where $N$ is the number of data points in the GP. As a result, recent works has proposed to use Sparse GPs, each prediction is made based an a small set of pseudo points reducing costs as compared to using the full dataset. This work proposes to use nearest neighbor GPs, each prediction is made based on using the nearest points in the dataset.

The authos

**Claims And Evidence:**

They authoprs claim their method has higher accuracy and quicker traiing times, I feel both of which are intuitive and justified by experiments.

**Essential References Not Discussed:**

[Longitudinal Variational Autoencoder Ramchandran et.al. AISTATS 2021](https://proceedings.mlr.press/v130/ramchandran21b.html)

**Experimental Designs Or Analyses:**

The paper contains many experiments from the literatire as well as some new ones.

- page 5, section 5.1, it is stated that GP-VAE Pearce is included but I don't see it anywhere, presumably in such a setting, the full expensive model of Pearce 2019 is the best case scenario and provides an upper bound on how any GP-VAE could perform in such a small scale use case?

I beleive the Longitudinal Variational Autoencoder should be added as a baseline for time series experiments [Longitudinal Variational Autoencoder Ramchandran et.al. AISTATS 2021](https://proceedings.mlr.press/v130/ramchandran21b.html)

**Methods And Evaluation Criteria:**

## Hierarchical Prior Approximation ELBO Derivation
I beleive I undertsand the intention of the approximation, however to me it seems the notation may have an error or two.
- The prior Equation 6 and 7, $p(Z|\textbf{w}) = \mathcal{N}(Z|0, D_W K_{XX}D_W)$ suggests that a masking vector $\textbf{w}\in \\{0, 1\\}^N$ sets some of the prior covariance rows/cols to 0, effectively the corresponding dimensions of $\textbf{Z}$ have a prior density that is a point mass at $z_i=0$, they become fixed constants and prior density is infinite everywhere (Note $det(D_wK_{XX}D_w)^{-1} = 0^{-1}=\infty$).
- (minor typo) Equation 8 gives the approximate posterior in Eq 8, assuming the masking vector $\textbf{w}$ is known whould also have mean 0 for the masked elements, $q(Z|\textbf{w}) = \mathcal{N}(Z|D_W\mu(Y), D_W K_{XX}D_W)$
- on P3, $L_{HPA}$ contains the term $q(Z|Y)$ which is not defined, though it is stated to be the same as Equation 4, the traditional VAE, presumably the unmasked approximate posterior. I beleive this to be an minor error, marginalizing
$\int_\textbf{w}q(Z|\textbf{w})p(\textbf{w})d\textbf{w}$ yields a mixture of gaussians, a point mass and the prior, (see [1], page 3, left col, bottom).
- on P3, $L_{HPA}$ contains the term $KL(q(Z|\textbf{w})||p(Z|\textbf{w}))$, if some dimensions of $Z$ are point masses then $p(Z|\textbf{w})$ both denstiies are infinite everywhere.

- Eq 10, the final ELBO $L_{HPA}$ seems intuitive and valid, the encoder should learn a good reconstruction as well as the neihbourhood should adhere to the local GP prior based on neighbour points. I believe a similar result can be acheived via a different root as the masking $\textbf{w}$ seems to make things a bit messy in my view and I beleive it does not have the same effect as simply removing dimensions (which is the desired goal and matcheas with Eq 10).

Presumably one could derive an ELBO f



[1] [Sparse within Sparse Gaussian Processes using Neighbor Information, Tran et.al.](https://arxiv.org/pdf/2011.05041)

**Other Comments Or Suggestions:**

Not at this time

**Other Strengths And Weaknesses:**

Overall I am positive about the paper. It is intuitive and simple and clean. My main concerns are cleaning up some maths

**Questions For Authors:**

- Can the derivation of Equation 10 be done by starting firectly from assuming we have a minibtach, and ignoring the hierachical approach?
- Are Pearce 2019 included in the results? If not can they be added
- Can the authors include the Longitudinal VAE where approriate?

**Relation To Broader Scientific Literature:**

This work nicely builds upon GP-VAE, SGP-VAE and provides an intuitive step forward showing expected performance gains.

**Theoretical Claims:**

There are no theoretical proofs. See Methods for comment on deriving the ELBO $L_{HPA}$

---

> ### Author Rebuttal · Authors · 2025-03-31
>
> We sincerely thank you for your valuable feedback. We provide a detailed response below.
>
> ---
> ### 1. Mathematical notations and clarification
>
> Conceptually, our approach closely aligns with the NNGPs proposed by [1], where a hierarchical mechanism enforces local sparsity on inducing variables. To draw a parallel, our GPVAE-HPA can be seen as applying such an idea to latent NNGPs with each “inducing variable” placed at each data point (similar to [2]). We then incorporate a decoder network to model the "likelihood" from $Z$ to $Y$ and an encoder network for amortised inference on $Y$, thereby forming our GPVAE architecture.
>
> We acknowledge that using $q(Z|Y)$ directly in $L_{HPA}$  (page 3) may cause confusion. We will replace it with $E_{p(w)}[q(Z|w,Y)]$, thus rewriting $L_{HPA}=E_{p(w)}[E_{q(Z|w,Y)}\log p(Y|Z)-KL[q(Z|w)|p(Z|w)]]$, which leads to an ELBO similar to Eq(10) of [1]. Here, $p(w)$ is any implicit distribution from which we can sample. In our implementation, $p(w)$ reflects a nearest-neighbour selection strategy akin to Eq (11) of [1]. We will clarify this connection with $L_{HPA}$ in the main text and Appendix B.1 and correct the minor notational issues in Eq(8) as $q(Z|w)=\mathcal{N}(Z|D_w\mu(Y),D_w\sigma^2(Y)D_w)$. We believe these revisions will both enhance the clarity of our derivation and highlight how GPVAE-HPA naturally builds upon established results for NNGPs.
>
> In our setup, the prior $p(Z|w)$ and the approximate posterior $q(Z|Y,w)$ in the KL share the same degenerate dimensions, meaning those components are effectively “turned off” in both distributions. It is reasonable to define their "KL contribution" as zero. Thus, the overall KL is effectively computed only on the non-degenerate subspace, ensuring the KL remains well-defined.
>
> ---
> ### 2. Longitudinal VAE (LVAE) as baseline
>
> The LVAE is also an inducing-point-based model, originally designed for longitudinal data with discrete “instance” IDs (e.g., a patient seen over multiple visits). This model embeds the instance ID into an additive kernel. Our experiments introduce *new* sequences at test time and do not explicitly provide IDs. Nevertheless, we have run additional experiments to address your request. We adapt LVAE by manually assigning unique IDs to each new sequence during testing. We presented the results of LVAEs with different numbers of inducing points $M$ from 10 random trials.
>
> The following table shows the performance of LVAEs on the Rotated MNIST *Missing Frame* experiment:
> |Model/Metric|NLL|RMSE|Training time(s/epoch)|
> |-|-|-|-|
> |LVAE-M10|73.887$\pm$0.748|5.578$\pm$0.022|28.919$\pm$0.253|
> |LVAE-M30|73.545$\pm$1.035|5.558$\pm$0.024|29.063$\pm$0.122|
> |GPVAE-HPA-H10(ours)|**71.264$\pm$1.263**|**5.521$\pm$0.046**|**9.819$\pm$0.064**|
> |GPVAE-SPA-H10(ours)|**69.538$\pm$0.664**|**5.412$\pm$0.026**|**9.358$\pm$ 0.083**|
>
> This table demonstrates that although LVAEs can get moderate results in terms of NLL and RMSE, our models consistently outperform LVAEs with fewer parameters and faster training.
>
> The following table illustrates the performance of LVAEs on the MuJoCo experiment:
> |Model/Metric|NLL|RMSE|Training time(s/epoch)|
> |-|-|-|-|
> |LVAE-M10|-0.003$\pm$0.316|0.175$\pm$0.021|**35.564$\pm$0.342**|
> |LVAE-M30|-0.814$\pm$0.201|0.107 $\pm$ 0.023|35.951$\pm$0.443|
> |GPVAE-HPA-H10(ours) |**-2.335$\pm$0.032**|**0.022$\pm$0.001**|37.000$\pm$3.793|
> |GPVAE-SPA-H10(ours) |**-1.715$\pm$0.159**|**0.038$\pm$0.007**|39.403$\pm$2.755|
>
> The table shows LVAEs exhibit substantially lower performance than our models with comparable training time.
>
> We will clarify the model differences in the revised manuscript and include relevant results in Section 5 of our paper and the appendix.
>
> ---
> ### 3. Including GPVAE-Pearce
>
> We included GPVAE-Pearce [3] as a *full GPVAE* baseline in Section 5.1. Specifically, we plot its performance with a dashed blue line in Figure 2(b) alongside GPVAE-Casale [4] to represent another “best-case” full-GP approach. Although both GPVAE-Pearce and GPVAE-Casale use fully correlated GP priors, they differ in how they construct their variational distributions—GPVAE-Pearce’s setup is more closely aligned with SVGPVAE [5], while GPVAE-Casale adopts a diagonal-encoder approach that matches ours. In our experiment, GPVAE-Casale slightly outperformed GPVAE-Pearce, so Figure 2(a) highlights GPVAE-Casale alone for clarity, whereas Figure 2(b) includes both baselines to show the overall trend.  We will clarify this point in the final revision to avoid confusion about whether GPVAE-Pearce was tested.
>
>
> [1] Sparse Within Sparse Gaussian Processes using Neighbor Information (ICML 2021)
>
> [2] Variational Nearest Neighbor Gaussian Process (ICML 2022)
>
> [3] The Gaussian Process Prior VAE for Interpretable Latent Dynamics from Pixels (AABI 2020)
>
> [4] Gaussian Process Prior Variational Autoencoders (NeurIPS 2018)
>
> [5] Scalable Gaussian Process Variational Autoencoders (AISTATS 2021)

---

### Decision · Program_Chairs · 2025-05-01

**Decision:**

Accept (poster)

**Comment:**

This paper proposes a method for improving the computational efficiency of Gaussian Process Variational Autoencoders (GP-VAEs) by incorporating the Nearest Neighbour Gaussian Process (NNGP) approach. The reviewers praised the paper for its clarity and the novelty of combining NNGPs with GP-VAEs, noting that it fills an obvious gap in the literature. The main criticisms centered around methodological choices, the selection of baselines, and the interpretation of results, with reviewers seeking further justification and clarification on these aspects.

The authors responded by addressing the concerns raised, providing additional explanations and justifications for their approach. The reviewers indicated that the authors' rebuttal has addressed their concerns, particularly regarding the methodological choices and experimental design, leading to a more positive assessment of the paper. Therefore, the reviewers unanimously agreed to accept the paper. We would still recommend that the authors take the reviewers' feedback into account when preparing the camera-ready version.